# MG-ViT: A Multi-Granularity Method for Compact and Efficient Vision Transformers

**Yu Zhang**[1]    **Yepeng Liu**[2]    **Duoqian Miao**[1]*    **Qi Zhang**[1]    **Yiwei Shi**[3]    **Liang Hu**[1]

[1]Tongji University    [2]University of Florida    [3]University of Bristol

## Abstract

Vision Transformer (ViT) faces obstacles in wide application due to its huge computational cost. Almost all existing studies on compressing ViT adopt the manner of splitting an image with a single granularity, with very few exploration of splitting an image with multi-granularity. As we know, important information often randomly concentrate in few regions of an image, necessitating multi-granularity attention allocation to an image. Enlightened by this, we introduce the multi-granularity strategy to compress ViT, which is simple but effective. We propose a two-stage multi-granularity framework, MG-ViT, to balance ViT's performance and computational cost. In single-granularity inference stage, an input image is split into a small number of patches for simple inference. If necessary, multi-granularity inference stage will be instigated, where the important patches are further subsplit into multi-finer-grained patches for subsequent inference. Moreover, prior studies on compression only for classification, while we extend the multi-granularity strategy to hierarchical ViT for downstream tasks such as detection and segmentation. Extensive experiments Prove the effectiveness of the multi-granularity strategy. For instance, on ImageNet, without any loss of performance, MG-ViT reduces 47% FLOPs of LV-ViT-S and 56% FLOPs of DeiT-S.

## 1 Introduction

Transformer [1] has gained tremendous achievements in computer vision. Despite the excellent performance of Convolutional Neural Networks (CNN) in computer vision tasks [2; 3; 4; 5], due to ViT's superior ability to capture global information and long-range interactions, ViT [6] outperforms CNN in various tasks [7; 8; 9; 10; 11]. However, the impressive performance of ViT comes at the cost of its huge computational overhead. Therefore, more researches focus on ViT compression for greater efficiency. Since the computational cost of ViT increases quadratically with the number of tokens, minimizing the number of tokens is crucial in compressing ViT.

Notable investigations have adopted various methods to compress ViT. DynamicViT [12] reduces token number by pruning redundant tokens, EViT [13] merges redundant tokens into one token. DVT [14] utilizes the particular and dynamic patch splitting manner (e.g., 4×4, 7×7, etc.) for each image, based on its complexity, rather than following the official manner of 14×14. Although DVT innovatively considers the differences in complexity between images, it still uses single granularity to split one image. As we all know, images often contain a lot of redundancy, with important semantic information randomly concentrated in a few regions. Therefore, it is imperative to consider the complexity and semantic density diversities across various regions of the image: different regions of an image should be assigned attention with multiple granularities. We initiate our exploration by observing the foundational task, classification, and we have 3 intriguing observations:

---

*Corresponding Author

**Observation 1**: Most images can be correctly recognized by splitting them into a small number of patches, as has been analogously observed in research [14]. Training a DeiT-S [15] with the $4 \times 4$ patch splitting manner can achieve 63.4% accuracy at a computational cost of 0.45G FLOPs. Although splitting an image to $7 \times 7$ patches and $14 \times 14$

Table 1: Accuracy and FLOPs of DeiT-S on ImageNet in different splitting manner.

| Manner | $14 \times 14$ | $7 \times 7$ | $4 \times 4$ |
|---|---|---|---|
| Accuracy | 79.8% | 73.2% | 63.4% |
| FLOPs | 4.60G | 1.10G | 0.45G |

patches can enhance the accuracy by 9.8% and 16.4%, the computation cost will also increase by $2.4\times$ and $10.2\times$. How to substantially improve the accuracy with a low growth rate of FLOPs is a subject worth contemplating.

**Observation 2**: The classification accuracy is mainly affected by the several critical tokens, and subsplitting patches into finer patches is more beneficial to classification. We split each image into $14 \times 14$ patches and rank tokens from high to low based on their class attention scores. The 196 tokens (patches) are equally divided into head, middle, and tail groups, and respectively input into DeiT-S to measure the accuracy of each group (Fig.1(a)). Then, subsplitting the patches in the head group expands the quantity to $4\times$ and $9\times$, their accuracy is measured (Fig.1(b)). It is illustrated that head tokens ensure the base of classification accuracy, and finer tokens lead to higher accuracy.

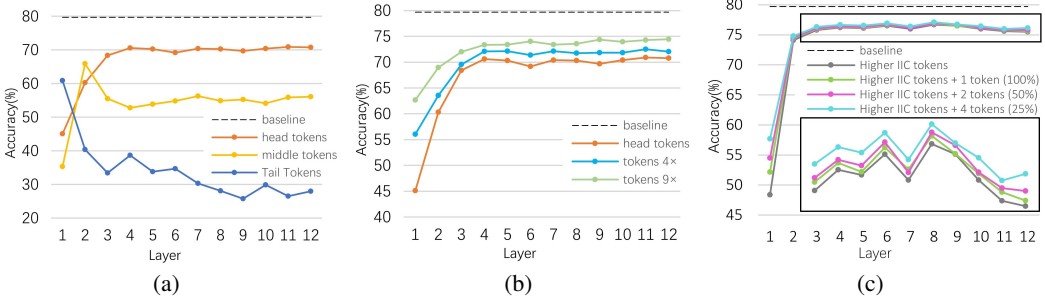

(a)  (b)  (c)

Figure 1: The classification accuracy of the tokens from different groups in each layer of DeiT-S. (c) 4 input groups: 1#: Only higher IIC tokens (top50%). 2#: Merging lower IIC tokens into one token and together with higher IIC tokens. 3#: According to CASR, merging every 50% of lower IIC tokens into one token (2 in total) and together with higher IIC tokens, 4#: According to CASR, merging every 25% of lower IIC tokens into one token (4 in total) and together with higher IIC tokens.

**Observation 3**: All information matters, and proper handling of tokens with low important information content (IIC) contributes to avoiding partial loss of accuracy. We treat the bottom 50% tokens as lower IIC tokens based on class attention score rank (CASR), and create 4 input groups for all tokens. The results in Fig.1(c) suggest that retaining lower IIC tokens can maintain accuracy. This is because lower IIC tokens contain complementary information to higher IIC tokens, which helps bridge the accuracy gap with the baseline. The dissimilarities among lower IIC tokens should be considered, necessitating a hierarchical processing. Performance improvement is from complementary information, certain lower IIC tokens make the higher contribution. If they are merged with other lower IIC tokens, the gain of performance improvement may be diluted.

Thus far, we can identify two sound reasons for utilizing the multi-granularity strategy to compress ViT. Firstly, as aforementioned, due to the distinctiveness of important information distribution in images, it is imperative to assign multi-granular attention to an image. Secondly, guided by our observations, we can consider multi-granularity as an intermediate state between full fine-grained and full coarse-grained. Full fine-grained stage represents stronger performance, while full coarse-grained stage represents less computation. Therefore, multi-granularity can naturally be used to balance performance and computational cost.

We propose MG-ViT: a multi-granularity framework for compressing ViT, which automatically splits an image into multi-granularity patches, as shown in Figure 2. MG-ViT consists of two parts: Single-Granularity Inference Stage (SGIS) and Multi-Granularity Inference Stage (MGIS), related to Observation 1. In SGIS, a small number of tokens are used for classification inference. Should a prediction confident enough ensue, the inference terminates immediately; otherwise, MGIS will be

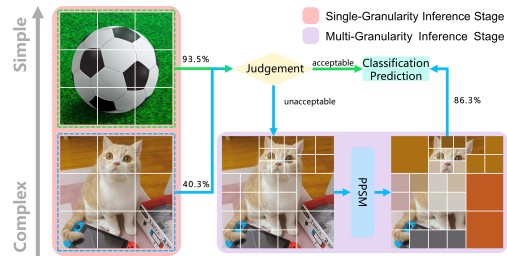

Figure 2: Example for MG-ViT.

triggered. In MGIS, We put forward three innovative methods. The first is multi-granularity patch subsplitting, related to Observation 2. All patches are divided into head, middle and tail groups based on CASR in SGIS, head and middle patches are subsplit, resulting in tokens with multiple information granularity that are input into ViT. The second is the three-way decision slimming, which can streamline ViT softly. Simply put, based on CASR, tokens are divided into three groups: positive, boundary, and negative. Boundary and negative tokens are respectively merged by different rules to reduce their numbers. The third is the token match-merge scheme for boundary tokens, related to Observation 3. The second and third methods are packaged as Plug-and-Play Slimming Module (PPSM) against the huge increase in computational cost caused by patch subsplitting. After PPSM, all tokens are fed into the next layer of ViT to continue inference.

Due to that ViT only outputs a single-scale feature representation and lacks the capability to handle multi-scale variations, it possesses significant deficiencies in downstream tasks. In order to adapt multi-granularity strategy to various downstream tasks, we extend the MG-ViT framework to hierarchical ViT. Unlike ViT, hierarchical ViT aggregates tokens within rectangular areas. However, patch subsplitting results in patches (tokens) with multiple scales interleaved in the image, making it challenging to seamlessly enclose them within regular rectangular areas. To address this issue, we introduce a proxy token of the same scale for each head or middle patch to represent itself. Additionally, each proxy token engages in attention calculations with its corresponding fine tokens or medium tokens to represent the subsplitting of the head or middle patch it stands for. Following the aggregation methodology of PVT [16], Hierarchical MG-ViT can perform downstream tasks.

MG-ViT is a dynamic and versatile framework that can be applied to most ViT models. We select LV-ViT [17] and DeiT [15] to assess the performance of MG-ViT on ImageNet [18]. The experiments demonstrate that MG-ViT can balance performance and computation cost greatly, thereby significantly improving the efficiency of ViT. Moreover, while ensuring performance, MG-ViT reduces 47% FLOPs of LV-ViT and 56% FLOPs of DeiT. We also conduct simple experiments in object detection and semantic segmentation on the MS-COCO [19] and ADE20K [20] datasets, respectively. The results show that Hierarchical MG-ViT effectively reduces computational overhead without significant performance loss, demonstrating the feasibility of extending multi-granularity strategy to the Hierarchical ViT structure.

## 2 Related Work

**Vision Transformer.** ViT [6] achieves a great breakthrough in the image classification task by redesigning the Transformer structure, completely igniting researchers' passion for exploring the ViT further. Later on, ViT is modified to hierarchical structure for various downstream tasks [21; 8; 22; 23].

**ViT and Hierarchical ViT.** ViT is a series of ViT models that adhere to the vanilla ViT design principles,such as DeiT [15] and LV-ViT[17]. They only output a single-scale feature representation and lack the capability to handle multi-scale variations, making them typically suitable for classification tasks. Hierarchical ViT is a series of ViT models that incorporate a hierarchical structure to aggregate tokens layer by layer, enabling them to better handle multi-scale information and well-suited for various downstream tasks. Swin-Transformer [24] and PVT [16; 25] are representative architectures of hierarchical ViT.

**ViT compression.** ViT and hierarchical ViT can both enhance efficiency. Hierarchical ViT primarily achieves efficiency by improving interaction rules among tokens, whereas ViT mainly relies on compression. ViT compression primarily aims to optimize ViT structures in a flexible and automated manner, developing a dynamic and efficient ViT where each image undergoes an individual computational process based on its unique features, all while minimizing computational costs without sacrificing performance. There are many classic ViT compression works, such as DynamicViT [12], PS-ViT [26], Evo-ViT [27], and so on [13; 14; 28; 29; 30; 31; 32; 33].

## 3 Multi-Granularity Vision Transformer

### 3.1 Overview

Figure 3 illustrates the overall framework of MG-ViT. For each image, SGIS must be executed. In SGIS, the image is split into a small number of patches. These 2D patches are embedded into 1D

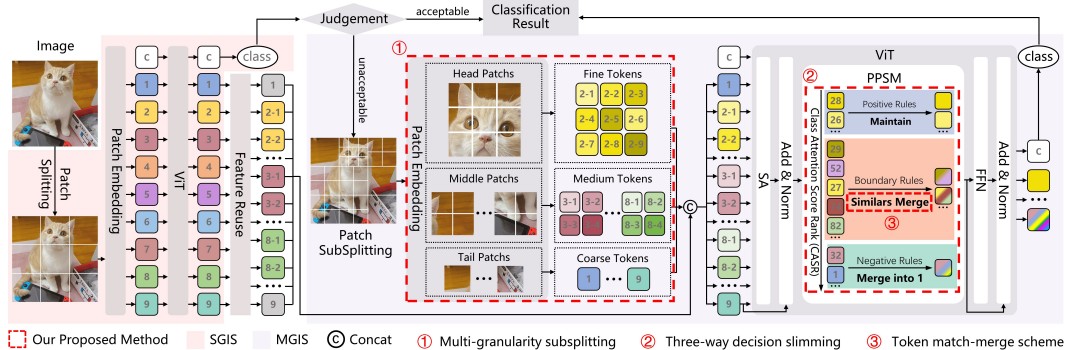

Figure 3: The overall framework of MG-ViT.

tokens and fed into ViT for straightforward inference. The obtained class prediction will be evaluated against predetermined criteria (threshold). If the class prediction is acceptable, the inference can be terminated immediately, and the class prediction can be output as the final classification result. If the class prediction is unacceptable, MGIS will be executed.

In MGIS, according to the IIC of each patch (referring to CASR), all patches are proportionally divided into three groups: head, middle and tail. Head and middle patches are further subsplit. As head patches with high IIC are the most crucial for classification, they require subsplitting with fine granularity, and eventually are subsplit into fine patches with fine-grained information. Middle patches with middle IIC are more crucial for classification but less than head patches. Whence, they require subsplitting with medium granularity, being subsplit into medium patches with medium-grained information. Tail patches, maintain their coarse granularity and serve as coarse patches. All patches undergo patch embedding and feature reuse to generate multi-granularity tokens which are input into ViT for inference, prediction, and output. Considering that patch subsplitting leads to a huge increase in computational cost, Plug-and-Play Slimming Module (PPSM) is introduced in certain layers of ViT to reduce computational cost. The three-way decision slimming lies here. According to CASR, we divide all tokens into three groups: positive, boundary and negative, aiming to realize soft slimming of ViT. Positive tokens remain unchanged. Among boundary tokens, similar tokens are matched and merged based on our proposed token match-marge scheme (more details in Section 3.3.2), resulting in fewer new tokens. Negative tokens are merged into one token by weights. In this way, the number of tokens is reduced, accomplishing ViT soft slimming.

## 3.2    Single-Granularity Inference Stage (SGIS)

In SGIS, the standard ViT is employed. Formally, for an input image $\aleph$, it is first split into patches $P = [p^1, p^2, ..., p^N]$ where $N$ is the number of patches. In patch embedding, patches are mapped into $C$-dimension token embeddings via linear projection. Additionally, a learnable token embedding $x^0$, referred to as a class token, is appended to the sequence of token embeddings to represent image $\aleph$ for class prediction. After position embedding $E_{pos}$, the input token sequence for ViT in SGIS is:

$$X = [x^0, x^1, ..., x^N] + E_{pos} \tag{1}$$

where $x^i \in \mathbb{R}^C$, present a token embedding of the $i$-th patch if $i > 0$, and $E_{pos} \in \mathbb{R}^{(N+1) \times C}$.

A ViT consists of $L$ layers, each layer comprises a self-attention (SA) module and a feed-forward network (FFN). In SA of the $l$-th layer, the self-attention *Attention*$(Q_l, K_l, V_l)$ is computed as follows:

$$\mathcal{A}_l = softmax(\frac{Q_l K_l^T}{\sqrt{C}}) = [a_l^0, a_l^1, ..., a_l^N], \qquad Attention(Q_l, K_l, V_l) = \mathcal{A}_l V_l, \tag{2}$$

where $Q_l, K_l, V_l \in \mathbb{R}^{(N+1) \times C}$, are query, key and value matrices of $l$-th layer respectively and obtained through linear projection for $X_{l-1}$. $\mathcal{A}_l$ is attention map of $l$-th layer, and the first row of attention map $a_l^0$ represent class attention in $l$-th layer.

Ulteriorly, the processes of SA and FFN in the $l$-th layer are as follows:

$$X'_{l-1} = SA(X_{l-1}) + X_{l-1}, \qquad X_l = FFN(X'_{l-1}) + X'_{l-1}. \tag{3}$$

After consecutive computation of $L$ layers, class token $x_L^0$ from the $L$-th layer is input into the classifier $\mathfrak{C}$ to obtain class prediction distribution $D^S$:

$$D^S = \mathfrak{C}(x_L^0) = [d_1^S, d_2^S, ..., d_M^S], \tag{4}$$

where $S$ indicates SGIS, $M$ denotes the class number, up to this point, the class prediction of image $\aleph$ is the largest entry of $D^S$, i.e., $argmax_j d_j^S$.

**Threshold for judgment.** If the value of $max_j d_j^S$ for input image $\aleph$ is large enough, it indicates that the class prediction in SGIS is acceptable, and the classification result can be generated. Therefore, the inference can be terminated immediately, saving computational cost. In order to use $max_j d_j^S$ to determine whether to terminate the inference, we introduce a threshold $\epsilon$: if $max_j d_j^S > \epsilon$, the inference is terminated immediately, and the class prediction is output as the classification result. If $max_j d_j^S < \epsilon$, The image is transitioned to MGIS for consecutive inference.

### 3.3 Multi-Granularity Inference Stage (MGIS)

#### 3.3.1 Generation of Multi-Granularity Tokens

To bridge the ViTs of two stages, we perform multi-granularity patch subsplitting and feature reuse to generate tokens with multi-grained information as the input for the ViT in MGIS.

**Multi-granularity patch subsplitting.** Since $max_j d_j^S$ of image $\aleph$ is less than $\epsilon$, further inference is required. As mentioned above, in MGIS, image $\aleph$ needs to be subsplit with multiple different granularities, and features of patches in SGIS need to be reused. These operations require to be instructed by IIC of patches (tokens) in SGIS, as patches with different IIC will be divided into different groups. Thus, the key now lies in how to identify the IIC of patches.

According to Equation 2, class token $y^0$ can be represented by class attention $a^0 \in \mathcal{A}$ as:

$$y^0 = softmax(\frac{q^0 K^T}{\sqrt{C}}) = a^0 \cdot V, \tag{5}$$

as we all known, class token $y^0$ represents the entire image $\aleph$ as input to produce the class prediction in classifier $\mathfrak{C}$. As $V = [v^0, v^1, ..., v^N]$, where $v^i$ is the value of the $i$-th token if $i > 0$. The class token $y^0$ is obtained by integrating the value of each token using $a^0$ as the corresponding weights. In other words, $a^0$ determines the amount of information from each token that enters the class token $y^0$ for classification. Therefore, we can use class attention $a^0$ to measure the contribution of each token to the class prediction, thence, we identify IIC according to CASR. However, if we only use CASR originating from the class attention of a certain layer to divide patches, patches of each group will be more random and unfixed, lacking overall consistency. Therefore, the flow of information across layers must also be considered. Thus, we come up with the global class attention instead of only using class attention of a certain layer to identify IIC of patches. The global class attention $\hat{a}_l$ in the $l$-th block is as following:

$$\hat{a}_l = \lambda \cdot \hat{a}_{l-1} + (1 - \lambda) \cdot a_l \tag{6}$$

where $\lambda = 0.98$. In Fig.1(a), we can observe that the classification accuracy of all groups exhibits intense fluctuation in shallow layers. It is evident that class attention is unstable in shallow layers. Therefore, the global class attention calculation starts from the 3-rd layer. We use CASR rooted from global class attention in the last layer $\hat{a}_L$ to identify IIC of patches.

According to $\hat{a}_L$, all patches are divided into three groups: head, middle, and tail. The numbers of head, tail and middle patches, denoted by $N_h$, $N_t$, and $N_m$ are given by:

$$N_h = \lfloor N \cdot r_h + 0.5 \rfloor, \qquad N_t = \lfloor N \cdot r_t + 0.5 \rfloor, \qquad N_m = N - N_h - N_t, \tag{7}$$

where $r_h$ and $r_t$ represent the number rate of head and tail patches respectively.

Head and middle patches need to be subsplit. Since head patches have high IIC, each is subsplit into $3 \times 3$ fine patches. Middle patches with middle IIC, each is subsplit into $2 \times 2$ medium patches. Each tail patch is considered as a coarse patch and remains unchanged. After patch subsplitting, the input token sequence is denoted as follows:

$$\bar{X} = [\bar{x}^0, \bar{x}^1, ..., \bar{x}^K] + \bar{E}_{pos}, \tag{8}$$

where $K$ represents the total number of patches after patch subsplitting.

**Feature reuse.** Patch subsplitting disrupts the integrity of head and middle patches, resulting in a lack of correlation among fine or medium patches which are severally subsplit from the same head or middle patch. To address this issue, we incorporate Feature Reuse module from DVT [14] to inject the original information of head or middle patches into new fine or medium patches. It is advantageous to enhance feature representation of each patch and strengthen the interconnectivity among them.

Figure 4 demonstrates the Feature Reuse module. Tokens (except class token) output by the ViT of SGIS are taken as input and divided based on their IIC. As head and middle patches need to be subsplit in MGIS, we individually upsample head and middle tokens, so that the number of new tokens obtained by upsampling is the same as the number of new patches obtained by subsplitting. Different from the module in DVT [14], considering that tail tokens remain unchanged in MGIS, we utilize zero padding to handle them. Finally, all tokens are flattened and sorted according to their position in image ℵ. The token sequence $X^{FR}$ output by Feature Reuse is obtained as follows:

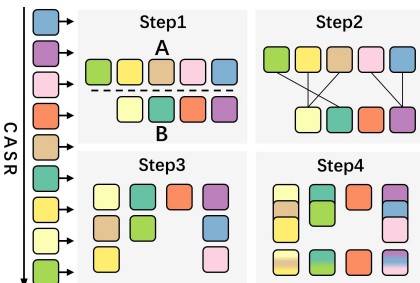

US: Unsampling   ZP: Zero Padding   F & S: Flatten & Sort

Figure 4: Feature Reuse module.

$$X^{FR} = FR(x_L^1, ..., x_L^N). \tag{9}$$

Specially, $X^{FR} \in \mathbb{R}^{N \times C'}$. Due to concatenation expanding the dimension of multi-granularity tokens to $2\times$, resulting in a $4\times$ increase in computational cost (detailed analysis in Appendix), we set a MLP in the module to realize dimension reduction: $MLP(\mathbb{R}^C \Rightarrow \mathbb{R}^{C'})$. The value of $C'$ is small, thereby increasing the dimension of multi-granularity tokens from $C$ to $C + C'$, but still much smaller than $2C$, which helps save computational cost.

The generated multi-granularity token sequence $\tilde{X} \in \mathbb{R}^{(N+1) \times (C'+C)}$, which is input to ViT in MGIS for inference, slimming and output, is denoted as:

$$\tilde{X} = Concat(\bar{X}, X^{FR}) = [\bar{x}^0, \tilde{x}^1, ..., \tilde{x}^N] \tag{10}$$

### 3.3.2 Plug-and-Play Slimming Module (PPSM)

Most existing ViT slimming methods, such as DynamicViT, PS-ViT, adopt token pruning, undermining the information integrity of the image and is not beneficial for classification. Therefore, we select token merging, which diminishes the number of tokens while preserving information.

**Three-way decision slimming.** Most existing ViT slimming methods belong to hard slimming, which divides tokens into non-redundant and redundant groups based on a certain criterion and reduces the number of redundant tokens to streamline ViT. Nevertheless, tokens in the middle part of CASR are arduous to distinguish as non-redundant or redundant during the inference. Mistakenly identifying non-redundant tokens as redundant and performing slimming will lead to a degradation of ViT's performance. Considering that three-way decision model establishes a buffer zone, i.e., boundary, between positive and negative, it can aid in more appropriately handling tokens situated in the middle position of CASR. Therefore, we employ three-way decision mechanism to realize soft slimming for ViT.

Three-way decision [34] is a decision mechanism in rough set [35; 36] that partitions a set into positive, negative, and boundary domains, with elements in each domain subject to positive, negative, and boundary rules, respectively. By setting the proportion of elements in positive and negative domains to $r_{pos}$ and $r_{neg}$, all tokens inputted into PPSM are divided into positive, negative and boundary domains. Positive tokens adhere to the positive rule: maintain unchanged. Boundary tokens adhere to boundary rule, which refers to the token match-merge scheme we proposed. Negative tokens adhere to the negative rule: all are merged into one token by weights.

Figure 5: The process of token match-merge.

**Token match-merge scheme.** As discussed in Observation 3, dissimilarities among lower IIC tokens should be considered, necessitating a hierarchical processing. Thus, we employ the token

match-merge scheme among boundary tokens to guide the match-merge process of similar boundary tokens, aiming to reduce the number of tokens while maintaining accuracy to the best possible extent. We measure the similarity between two tokens by computing the cosine similarity of their value vectors. $v^a, v^b \in \mathbb{R}^C$, present the value vectors of tokens $a$ and $b$, and the cosine similarity $cos(\theta)$ is expressed as follows:

$$cos(\theta) = \frac{\sum_{i=1}^{C+C'}(v_i^a \times v_i^b)}{\sqrt{\sum_{i=1}^{C+C'}(v_i^a)^2} \times \sqrt{\sum_{i=1}^{C+C'}(v_i^b)^2}} \tag{11}$$

The token match-merge scheme is demonstrated in Figure 5. Step1. Divide tokens at odd positions in CASR into Group A and those at even positions into Group B. Step2. Find the most similar token in set B for each token in set A by calculating cosine similarity. Step3. Put similar tokens together to complete the match. Step4. Merge the similar tokens by weights to reduce the number of tokens.

### 3.4 Training Mehtod

During the training of MG-ViT, setting $\epsilon = 1$ makes MGIS be executed for every input image. Like knowledge distillation [37], our goal is to make the outputs of SGIS more similar to MGIS. Thus ViT in SGIS can have a stronger ability in classification. The training loss about ground-truth label $gt$ is:

$$loss = CE(D^M, gt) + KL(D^S, D^M), \tag{12}$$

where $CE(\cdot, \cdot)$ and $KL(\cdot, \cdot)$ represent cross entropy loss and Kullback-Leibler divergence.

## 4  Hierachical Multi-Granularity Vision Transformer

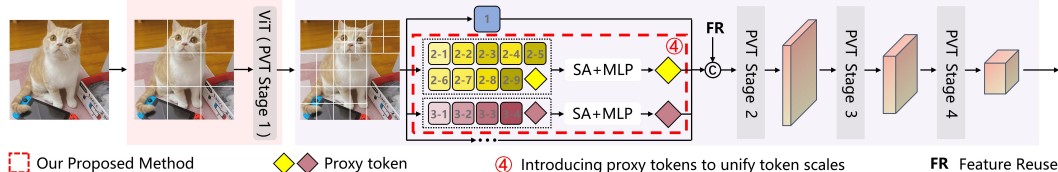

Figure 6: The overall framework of Hierarchical MG-ViT.

### 4.1 Key Issue of Extension

Almost all classic ViT compression works, such as Dynamic ViT [12], PS-ViT [26], EViT [13], only focus on classification. Why not extend their methods to hierachical ViT to execute downstream tasks like detection and segmentation? We analyze two reasons. The first reason is structural limitation. For instance, plain ViT only provides a single-scale feature representation, lacks the ability to handle multi-scale variation, and lacks image-related priors. While there are a few works that still employ plain ViT for downstream tasks, such as ViTDet [38], ViT-Adapter [39], SegViT [40; 41], they achieve this by integrating well-designed modules, which could be considered as individual research contributions. The second reason is method challenges. Most classic ViT compressions, as mentioned above, heavily rely on pruning or merging, which can be challenging to adapt to downstream tasks. Although we also employ pruning and merging methods, they serve as auxiliary means to reduce computational costs. The core of this paper - multi-granularity strategy - can be applied in hierarchical ViT for downstream tasks.

The key to extending multi-granularity strategy to hierarchical ViT lies in addressing the non-uniform issue of multi-grained patch scales. Subsplitting head and middle patches into $3 \times 3$ fine patches and $2 \times 2$ medium patches results in patches with multiple scales interweaving in an image, making it difficult to seamlessly enclose patches in a conventional window when using Swin Transformer. Therefore, in this paper, we propose proxy tokens to unify scales.

### 4.2 Introducing Proxy Tokens for Unifying Scales

Similar to MG-ViT, Hierarchical MG-ViT also employs a two-stage framework. In SGIS, 2D image $\aleph \in \mathbb{R}^{H \times W}$ is split into patches $P \in \mathbb{R}^{\frac{H}{s} \times \frac{W}{s} \times s^2}$, where $(H, W)$ donates the resolution of image $\aleph$,

and $s$ presents the patch size. The token sequence generated by $P$ embedding is:

$$T_{ori} = [t_{ori}^1, t_{ori}^2, ..., t_{ori}^{\frac{H}{s} \times \frac{W}{s}}], \tag{13}$$

where $T_{ori} \in \mathbb{R}^{\frac{HW}{s^2} \times C_1}$, $ori$ is a simplified representation of "origin". IIC of $P$ is identified in SGIS.

In MGIS, we assume that $P^i \in \mathbb{R}^{s \times s}$ is a patch that requires subsplitting, which will be subsplit into $\ell \times \ell$ fine patches:

$$P^i = [p^{i,1}, p^{i,2}, ..., p^{i,\ell^2}], \tag{14}$$

where $\ell \times \ell$ is the manner of subsplitting ($3 \times 3$ or $2 \times 2$), $p^{i,j}$ present the $j$-th fine or medium patch subsplit from the $i$-th head or middle patch if $i > 0$, and $j > 0$.

Then the finer token sequence $T_{ori}^i \in \mathbb{R}^{\ell^2 \times C_1}$ originated from the embedding of these finer patches is denoted as:

$$T_{ori}^i = [t_{ori}^{i,1}, t_{ori}^{i,2}, ..., t_{ori}^{i,\ell^2}]. \tag{15}$$

We introduce a proxy token $t_{pro}^i \in \mathbb{R}^{C_1}$ for this group of finer tokens to compose the new sequence $T^i = [T_{ori}^i, t_{pro}^i]$. The computing procedure of a Transformer layer for this group can be summarized as follows:

$$T^{i'} = SA(T^i) + T^i, \qquad T^{i'} = MLP(T^{i'}) + T^{i'}. \tag{16}$$

To generalization, We provide an optional proxy token set $T_{pro}$ and its mask vector $m$ for each image, where $T_{pro} \in \mathbb{R}^{\frac{HW}{s^2} \times C_1}$ and $m \in \mathbb{R}^{\frac{HW}{s^2}}$. Thus, the unified scale token sequence $T'$ before feature reuse can be denoted as:

$$T' = (-m) \times T_{ori} + m \times T_{pro}'. \tag{17}$$

After feature reuse, the token sequence is input into the pyramid-structured hierarchical ViT, through PVT stages 2, 3, and 4, feature maps reshape to $\frac{H}{2s} \times \frac{W}{2s} \times C_2$, $\frac{H}{4s} \times \frac{W}{4s} \times C_3$ and $\frac{H}{8s} \times \frac{W}{8s} \times C_4$. In this way, various downstream tasks can be performed.

## 4.3 Architectural Details

We develop two variants of Hierarchical MG-ViT based on PVT-Small: Hierarchical MG-ViT-A and Hierarchical MG-ViT-R. Hierarchical MG-ViT-A adds a Transformer layer with proxy tokens between stage 1 and stage 2 of PVT-Small. Hierarchical MG-ViT-R maintains the same capacity as PVT-Small but replaces one of the original Transformer layers in stage 1 with a Transformer layer introducing proxy tokens. Please refer to the appendix for architectural details.

## 4.4 Discussions

**Performance upgrade.** Since our proposed method is plug-and-play and does not significantly alter the structure of the hierarchical ViT, methods aimed at improving the performance of hierarchical ViT, such as using relative position biases and overlapping patch embedding, are all effective. However, we haven't optimize Hierarchical MG-ViT for the best performance, as our goal was to explore the potential of extending the multi-granularity strategy from compressing plain ViT to hierarchical ViT for various downstream tasks. In the future, we will delve deeper into performance enhancements for Hierarchical MG-ViT.

**Plain ViT $vs.$ hierarchical ViT.** Which ViT structure is superior? Plain ViT inherently has limitations when it comes to handling dense tasks. In order to utilize a general ViT for a variety of tasks, many new ViTs are designed based on hierarchical structures. However, we do not consider hierarchical ViT to be superior; both plain ViT and hierarchical ViT have their strengths and weaknesses. As for detection tasks, a multitude of ViTs built upon the hierarchical structure continue to remain the most competitively poised model. However, models like ViTDet [38], which are based on plain ViT, also demonstrate that hierarchical ViTs such as Swin-T and PVT are not the only efficient ways to accomplish visual tasks. As for segmentation tasks, what surprised us is that SegViTv2 [41], built upon the Plain ViT, has achieved state-of-the-art results. As for classification tasks, a variety of plain ViT models, with MG-ViT as a representative, exhibit a big performance advantage over hierarchical ViT. Moreover, multi-granularity strategy boost performance more effectively in plain structures than in hierarchical structures. This is because hierarchical ViT involves token merging, which

significantly diminishes the advantages of multi-granularity in feature representation. Regarding the question of which ViT structure is superior, we believe that it necessitates deliberation in the context of varying tasks and distinct constraints.

## 5 Experiments

### 5.1 Backbones, Datasets and Evaluation Metrics.

We built MG-ViT based on DeiT-S and LV-ViT-S and evaluated its performance on the benchmark ImageNet [18]. To quantitatively compare performance, we report the Top-1 accuracy (Acc.), the number of floating-point operations (FLOPs), and throughput (TP). Moreover, we developed Hierarchical MG-ViT based on PVT-Small and evaluated its performance in object detection and semantic segmentation on the benchmark datasets MS-COCO2017 [19] and ADE20K [20]. We reported metrics such as box average precision ($AP^b$), mean intersection over union (mIoU) and the number of parameters (#Params). All metrics are measured on a single NVIDIA RTX 3090 GPU.

### 5.2 Main Results

**Comparison to backbones.** Table 2 presents the comparison results between MG-ViT and backbones with different values of threshold $\epsilon$. It can be observed that Without any performance degradation, the computational cost of MG-ViT is significantly reduced. Moreover, when all images execute MGIS, the performance of ViT is improved, which can be attributed to subsplitting patches with multiple different granularities.

Table 2: Comparison between MG-ViT and backbones

| Model | $\epsilon$ | Acc.(%) | FLOPs(G) | TP(img./s) |
|---|---|---|---|---|
| DeiT-S | - | 79.8 | 4.6 | 1341 |
| MG-ViT | 0.49 | 79.8(+0.0) | 2.0(-56%) | 2404(+1.79×) |
| MG-ViT | 0.73 | 80.8(+1.0) | 2.4(-48%) | 2054(+1.53×) |
| MG-ViT | 1 | 81.0(+1.2) | 3.9(-15%) | 1591(+1.11 ×) |
| LV-ViT-S | - | 83.3 | 6.6 | 989 |
| MG-ViT | 0.60 | 83.3(+0.0) | 3.5(-47%) | 1749(+1.77×) |
| MG-ViT | 0.73 | 83.7(+0.4) | 3.8(-42%) | 1683(+1.70×) |
| MG-ViT | 1 | 83.8(+0.5) | 5.6(-15%) | 1233(+1.25×) |

**Comparison to baselines.** In Figure 7, we compare MG-ViT with 3 baselines: DynamicViT, EViT, and DVT. These models are chosen as representatives of 3 type methods: token pruning, token merging, and dynamic design of token format. With the same setting, MG-ViT outperforms baselines.

**Comparison to other ViT compressing methods.** In order to demonstrate the effectiveness of our dynamic framework, MG-ViT, in compressing ViT, we present a comparison of MG-ViT with various ViT compressing methods in Table 3. We visualize their accuracy, FLOPs, and throughput in Appendix. The results show that our method for compressing ViT based on the multi-granularity strategy, MG-ViT, is highly competitive when using DeiT-S and LV-ViT-S as backbones.

Table 3: Comparison between MG-ViT and other ViT compressing methods.

| LV-ViT-S | | | | DeiT-S | | | |
|---|---|---|---|---|---|---|---|
| Model | FLOPs(G) | Acc.(%) | TP(img/s) | Model | FLOPs(G) | Acc.(%) | TP(img/s) |
| Baseline[17] | 6.6 | 83.3 | 989 | Baseline[15] | 4.6 | 79.8 | 1341 |
| PS-ViT[26] | 4.7 | 82.4 | - | IA-RED²[29] | 3.3 | 79.1 | 1597 |
| eTPS[42] | **3.8** | 82.5 | 1665 | DVT[14] | **2.4** | 79.3 | 1485 |
| EViT[13] | 4.7 | 83.0 | 1447 | DynamicViT[12] | 2.9 | 79.3 | 1774 |
| DynamicViT[12] | 4.6 | 83.0 | 1302 | Evo-ViT[27] | 2.9 | 79.4 | 1863 |
| SPViT[32] | 4.3 | 83.1 | 1518 | ATS[43] | 2.9 | 79.7 | 1531 |
| SiT[30] | 4.0 | 83.4 | 1280 | STViT[33] | 3.2 | 80.6 | 1928 |
| CF-ViT[28] | 4.0 | 83.5 | **1711** | CF-ViT[28] | 2.6 | 80.7 | **2096** |
| **MG-ViT** | **3.8** | **83.7** | 1683 | **MG-ViT** | **2.4** | **80.8** | 2054 |

**Comparison to other efficient ViTs.** In Figure 8, we compare MG-ViT and various efficient ViTs [44; 45; 46; 47; 48; 49; 50; 51; 52; 53; 25; 54; 55; 56; 57; 58; 59; 60; 61; 62; 63; 64]. It can be observed that MG-ViT is competitive even among different backbones and other ViT variants.

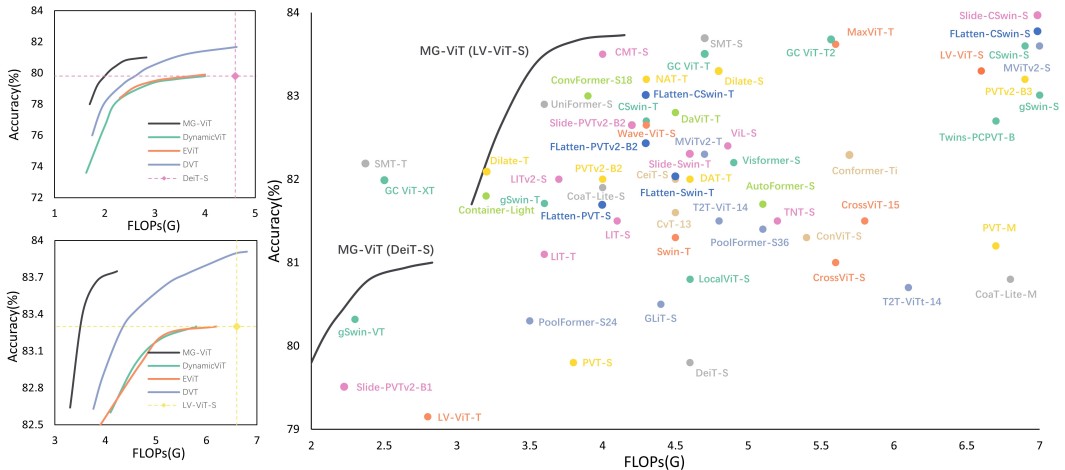

Figure 7: Comparison with baselines.

Figure 8: Comparison MG-ViT between and static efficient ViT methods.

**Object detection and semantic segmentation.** We adopt the same implementation details as PVT-Small and conduct preliminary experiments on Hierarchical MG-ViT's performance in object detection and semantic segmentation tasks to represent its capability for executing downstream tasks. As shown in Table 4 and Table 5, extending the multi-granularity strategy to the hierarchical structure is beneficial for enhancing ViT's performance and serves as a nice attempt to slim hierarchical ViT.

Table 4: Object detection with Mask R-CNN $1\times$ schedule.

| Method | #Params(M) | FLOPs(G) | $AP^b$ | $AP^b_{50}$ | $AP^b_{75}$ |
|---|---|---|---|---|---|
| PVT-Small [16] | 44.1 | 245 | 40.4 | 62.9 | 43.8 |
| **Hierarchical MG-ViT-A** | 46.5 | 258 | **40.9** | 63.2 | 44.2 |
| **Hierarchical MG-ViT-R** | **41.9** | **227** | 40.3 | 62.7 | 43.6 |

Table 5: semantic segmentation with semantic FPN head.

| Method | #Params(M) | FLOPs(G) | mIoU |
|---|---|---|---|
| PVT-Small [16] | 28.2 | 44.5 | 39.8 |
| **Hierarchical MG-ViT-A** | 29.3 | 46.4 | **40.1** |
| **Hierarchical MG-ViT-R** | **27.1** | **41.3** | 39.7 |

# 6   Conclusion

MG-ViT, the two-stage dynamic framework based on the strategy we proposed, achieves a well-balanced trade-off between performance and computational cost. Within this novel framework, we contribute four new methods. Firstly, we subsplit patches with multiple different granularities, resulting in generating multi-granularity tokens with various levels of information, which enhances the capacity of ViT in feature representation, leading to performance improvement. Secondly, we introduce a three-way decision mechanism to achieve soft slimming of ViT. Thirdly, we propose the token match-merge scheme to guide the match-merge of boundary tokens, reducing their quantity. The second and third methods are reflected in our proposed PPSM. Fourthly, we extend multi-granularity strategy to hierarchical ViT for various downstream tasks, demonstrating that our method can also be applied to hierarchical structures for compressing, and we will explore this further in the future.

# Acknowledgement

This work is supported by the National Key Research and Development Program of China (No. 2022YFB3104700), the National Natural Science Foundation of China (No. 61976158, No. 62376198, No. 62006172, No. 62076182, No. 62163016, and No. 62106091), the Jiangxi "Double Thousand Plan", the Jiangxi Provincial Natural Science Foundation (No. 20212ACB202001) and the Shandong Provincial Natural Science Foundation (No. ZR2021MF054). The authors would like to thank Jun Wang, Yanping Li, Qixian Zhang, Zhiheng Wu for inspirational and feasible suggestions.

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

# 7  Appendix

## 7.1  Hierarchical MG-ViT Architectural Details

We present two variants of the Hierarchical MG-ViT designed based on PVT-Small, as shown in Table 6. The relevant hyper parameters are listed as follows: $P_i$: the patch size of Stage $i$; $C_i$: the channel number of the output of Stage $i$; $R_i$: the reduction ratio of the spatial-reduction attention in Stage $i$; $N_i$: the head number of the spatial-reduction attention in Stage $i$; $E_i$: the expansion ratio of the feed-forward layer in Stage $i$; $H_i$: the head number of the multi-head attention in Stage $i$.

Table 6: Architectural details of Hierarchical MG-ViT and PVT-Small

| | Output Size | Layer Name | PVT-Small | Hierarchical MG-ViT-A | Hierarchical MG-ViT-R |
|---|---|---|---|---|---|
| Stage 1 | $\frac{H}{4} \times \frac{H}{4}$ | Patch Embedding | $P_1 = 4; C_1 = 64$ | | |
| | | Transformer Encoder | $\begin{bmatrix} R_1 = 8 \\ N_1 = 1 \\ E_1 = 8 \end{bmatrix} \times 3$ | $\begin{bmatrix} R_1 = 8 \\ N_1 = 1 \\ E_1 = 8 \end{bmatrix} \times 3$ | $\begin{bmatrix} R_1 = 8 \\ N_1 = 1 \\ E_1 = 8 \end{bmatrix} \times 2$ |
| Stage MG | $\frac{H}{4} \times \frac{H}{4}$ | MG Transformer | - | $\begin{bmatrix} H_{MG} = 8 \\ E_{MG} = 8 \end{bmatrix} \times 1$ | $\begin{bmatrix} H_{MG} = 8 \\ E_{MG} = 8 \end{bmatrix} \times 1$ |
| Stage 2 | $\frac{H}{8} \times \frac{H}{8}$ | Patch Embedding | $P_2 = 2; C_2 = 128$ | | |
| | | Transformer Encoder | $\begin{bmatrix} R_2 = 4 \\ N_2 = 2 \\ E_2 = 8 \end{bmatrix} \times 3$ | $\begin{bmatrix} R_2 = 4 \\ N_2 = 2 \\ E_2 = 8 \end{bmatrix} \times 3$ | $\begin{bmatrix} R_2 = 4 \\ N_2 = 2 \\ E_2 = 8 \end{bmatrix} \times 3$ |
| Stage 3 | $\frac{H}{16} \times \frac{H}{16}$ | Patch Embedding | $P_3 = 2; C_3 = 320$ | | |
| | | Transformer Encoder | $\begin{bmatrix} R_3 = 2 \\ N_3 = 5 \\ E_3 = 4 \end{bmatrix} \times 6$ | $\begin{bmatrix} R_3 = 2 \\ N_3 = 5 \\ E_3 = 4 \end{bmatrix} \times 6$ | $\begin{bmatrix} R_3 = 2 \\ N_3 = 5 \\ E_3 = 4 \end{bmatrix} \times 6$ |
| Stage 4 | $\frac{H}{32} \times \frac{H}{32}$ | Patch Embedding | $P_4 = 2; C_4 = 512$ | | |
| | | Transformer Encoder | $\begin{bmatrix} R_4 = 1 \\ N_4 = 8 \\ E_4 = 4 \end{bmatrix} \times 3$ | $\begin{bmatrix} R_4 = 1 \\ N_4 = 8 \\ E_4 = 4 \end{bmatrix} \times 3$ | $\begin{bmatrix} R_4 = 1 \\ N_4 = 8 \\ E_4 = 4 \end{bmatrix} \times 3$ |

## 7.2  Implementation Details

The resolution of input images in our experiments is $224 \times 224$. In SGIS, we split each image into $7 \times 7$ patches. $r_h$ and $r_t$ are set to 0.1 and 0.4, respectively. Therefore, each image is subsplit into 161 patches in MGIS. For DeiT-S, we inserted a total of three PPSM modules in the 3rd, 7th, and 10th layers. Similarly, for LV-ViT-S, we inserted a total of four PPSM modules in the 4th, 7th, 9th, and 12th layers. For conducting the training process, we set the batch size to 256 and use AdamW optimizer to train all models for 300 epochs.

In terms of training strategies and optimization methods, our training closely follows the original methods proposed in the studies conducted by DeiT and LV-ViT, but do not employ any knowledge distillation algorithms in our experiments. Throughout the training phase, we observed the impact on convergence when subsplitting was applied solely to the head and middle patches. To enhance the speed of model convergence, we adopted subsplitting specifically targeting the head and middle patches for the initial 200 epochs. Subsequently, for the following 100 epochs, we applied subsplitting to the head and tail patches. The training was performed on a workstation equipped with 8 RTX 3090 GPUs.

## 7.3 Visualization Result

In Figure 9, we specifically demonstrate the images that can be well recognized by MG-ViT (LV-ViT-S) in SGIS and MGIS. As observed, the images on the left are "simple," usually occupying a significant portion of the image pixels, with typically complete and clear contours. While the images on the right are "complex", containing intricate scenes and non-prominent objects or only include a small part of the objects, requiring multi-granularity representations through additional tokens.

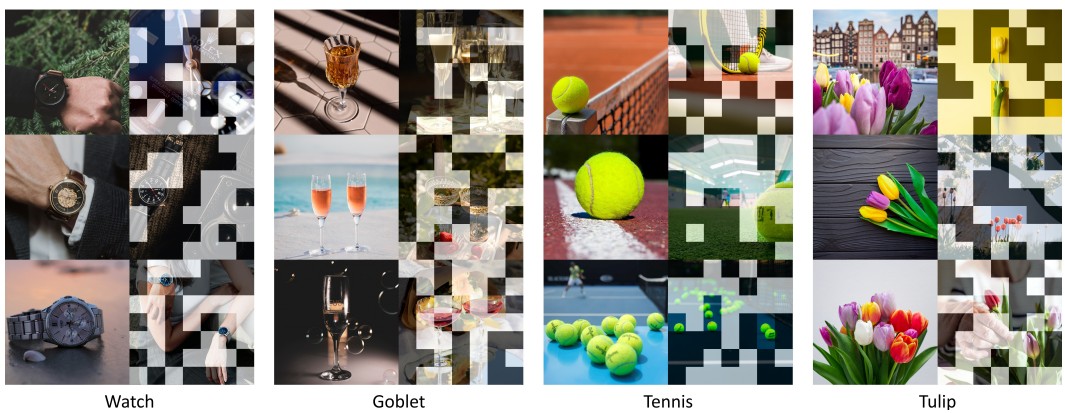

Watch      Goblet      Tennis      Tulip

Figure 9: Example of images well recognized in SGIS and MGIS. To provide clarity, we use unfilled, semi-white, and semi-black boxes to represent the head, middle, and tail patches in MGIS, respectively.

Figure 10 visualizes the token slimming comparison between token merging (MG-ViT) and token pruning (DynamicViT). The result shows, the information loss caused by token pruning may result in incorrect slimming decisions, leading to a decrease in classification accuracy. However, the adoption of token merging in PPSM alleviates this situation.

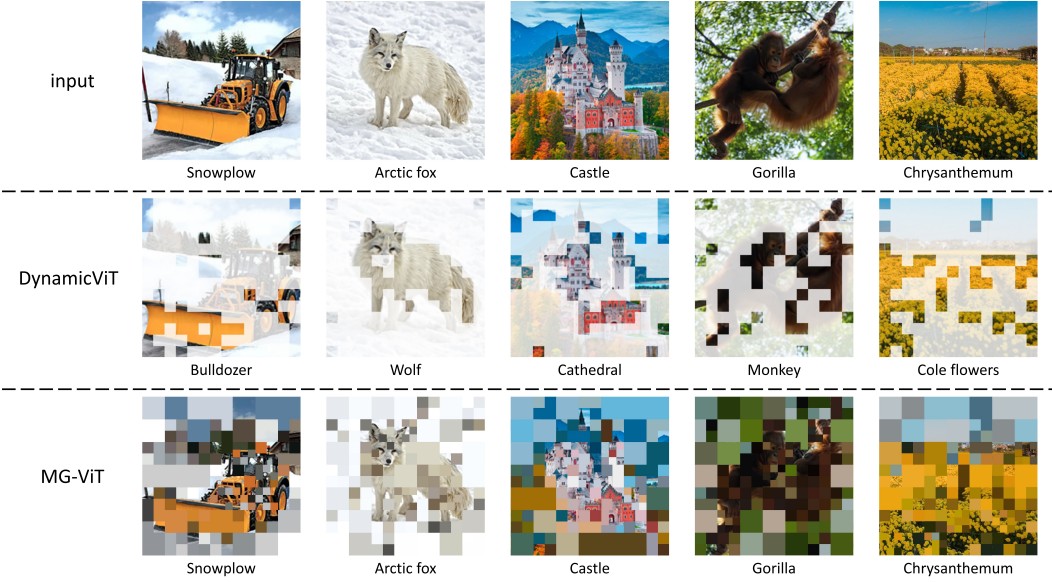

Figure 10: Visual comparison between token merging and pruning. The boxes with the same color represent a merged token.

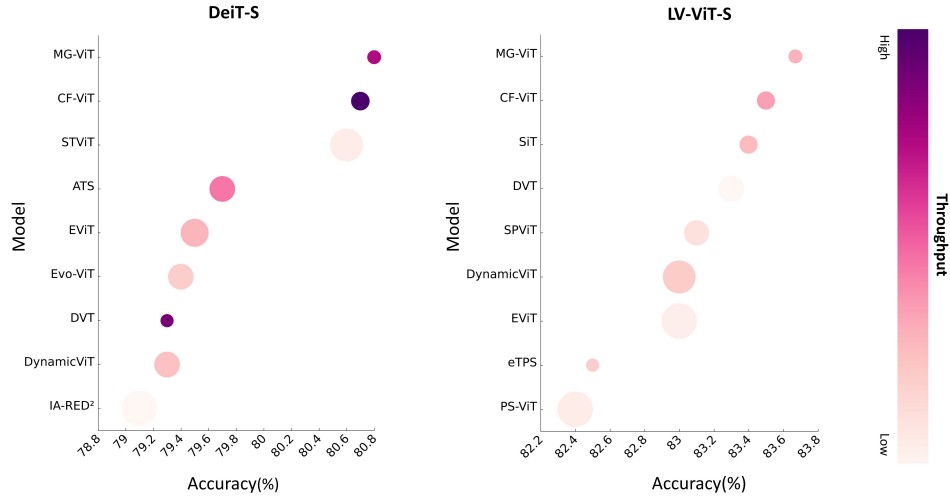

Figure 11: Comparison between MG-ViT and existing efficient ViT methods. The size of ● represents the computational cost required for each image (FLOPs).It can be observed that MG-ViT achieves a good balance in terms of accuracy, FLOPs, and throughput, demonstrating its overall competitiveness.

Table 8: Ablation experiment on global class attention.

| Scheme | Acc.(%) | |
| --- | --- | --- |
| | SGIS | MGIS |
| Random division | 75.0 | 78.9 |
| Random CASR | 75.3 | 79.8 |
| Last CASR | 75.5 | 80.5 |
| Ours | 75.6 | 80.8 |

Table 9: Ablation experiment on Feature Reuse module.

| Scheme | Acc.(%) | |
| --- | --- | --- |
| | SGIS | MGIS |
| w/o Feature Reuse | 75.0 | 79.9 |
| Ours + Class Token | 75.4 | 80.1 |
| w/o Zero-Padding | 75.5 | 80.7 |
| Concat→add | 75.4 | 80.7 |
| Ours | 75.6 | 80.8 |

## 7.4 More Experiments and Ablation Analysis

**Necessity of global class attention.** We designed two experiments based on backbone DeiT-S to investigate the necessity of global class attention.

Table 7: Accuracy with different values of $\lambda$.

| $\lambda$ | 0 | 0.5 | 0.9 | 0.95 | 0.98 | 0.99 | 0.999 |
| --- | --- | --- | --- | --- | --- | --- | --- |
| Acc.(%) | 80.3 | 80.5 | 80.7 | 80.7 | 80.8 | 80.7 | 80.7 |

Table 8 shows the gain of the proposed global attention class on model performance, compared to random division, which involves randomly dividing patches into three groups, random CASR, which divides patches according to CASR in a random layer of ViT, and last CASR, which divides patches based on CASR in the last layer. As we can see, the global attention class achieves the best performance improvement. Table 7 compares the contribution of the global attention class to model performance with different values of $\lambda$. When $\lambda = 0$, it represents the last CASR scheme. We set the default value of $\lambda$ as 0.98.

**Applicability of Feature Reuse.** We introduce the Feature Reuse module from DVT[14] to enhance the feature representation of patches, and also explore its applicability in our model. Specially, we do not reuse the class tokens and apply zero padding to the tail tokens (i.e., not reusing the information from tail tokens). Table9 demonstrates the effective integration of the Feature Reuse module into our model, and indicates the effectiveness of our improvements.

**Effect of token match-merge scheme.** We conduct ablation experiments to investigate the effort of our proposed token match-merge scheme and establish three control schemes for comparison: token pruning, token clustering (K-means), and merging tokens into one. Especially, by employing our token match-merge scheme on boundary tokens, we achieved the best trade-off between accuracy and speed, as shown in Table 10. This result also supports the rationality of merging all negative tokens into one, because we need to process the tokens with low IIC in the most computationally

| Table 10: Ablation experiment on the token match-marge scheme. | | |
| --- | --- | --- |
| Scheme | Acc.(%) | SPD(img./s) |
| Token pruning | 79.5 | 188 |
| Token clustering | 79.8 | 173 |
| Merging tokens into 1 | 80.4 | 183 |
| Match-marge (ours) | 80.8 | 180 |

| Table 11: Comparison among various merging approaches. | |
| --- | --- |
| Approach | Acc.(%) |
| Keep one | 80.3 |
| Max pool | 80.4 |
| Avg pool | 80.6 |
| weighted avg(ours) | 80.8 |

economical manner. Regarding the approach of generating new tokens by merging, the results in Table 11 demonstrate that the weighted average method performs best in maintaining accuracy.

**Token merging** *vs.* **token pruning.** Table 12 displays the performance of token pruning and token merging under two types of slimming, namely the hard slimming type (dividing tokens into two groups) and the soft slimming type (dividing tokens into three groups). It is observed that utilizing three-way decision mechanism for token merging (i.e., our proposed PPSM) achieves the best performance. This is attributed to assigning tokens located in the middle part of CASR to the boundary domain within three-way decision model, where the tokens adhere to boundary rules, resembling a "delayed processing".

| Table 12: Comparison between hard and soft slimming. | | |
| --- | --- | --- |
| Type | Method | Acc.(%) |
| Hard | Pruning | 79.8 |
| | Merging | 80.3 |
| Soft | Pruning | 80.2 |
| | Merging | 80.8 |

Table 13: Experiments fine-tuned and evaluated with a resolution of 384.

| Model | $\epsilon$ | Acc.(%) | FLOPs(G) | TP(img./s) |
| --- | --- | --- | --- | --- |
| LV-ViT-S $^{\uparrow 384}$ | - | 84.4 | 22.2 | 303 |
| MG-ViT $^{\uparrow 384}$ | 0.66 | 84.4(+0.0) | 10.7(−52%) | 515(+1.70×) |
| MG-ViT $^{\uparrow 384}$ | 1 | 85.5(+0.7) | 16.9(−24%) | 345(+1.14×) |

**Effectiveness of PPSM.** We integrate the two methods, three-way decision slimming and token match-merge scheme, into our proposed PPSM module. Table 14 compares the performance of MG-ViT (PPSM) with existing methods of hard slimming and soft slimming, as well as the performance between MG-ViT with and without PPSM. The results demonstrate that our proposed PPSM module effectively streamlines the model while maintaining performance.

Table 14: Comparison between MG-ViT and the existing soft and hard slimming methods for ViT

| Type | Method | Model | Acc.(%) | FLOPs(G) |
| --- | --- | --- | --- | --- |
| Origin | - | MG-ViT(w/o PPSM) | 80.8 | 3.7 |
| Baseline | - | DeiT | 79.8 | 4.6 |
| Hard slimming | Pruning | PS-ViT | 79.8(−0.0) | 3.8(−1.2) |
| | | DynamicViT | 79.3(−0.5) | 2.9(−1.7) |
| | | IA-RED² | 79.1(−0.7) | 3.3(−1.3) |
| | Merging | EViT | 79.5(−0.3) | 3.0(−1.6) |
| Soft slimming | Pruning | SiT | 79.8(−0.0) | 2.3(−2.3) |
| | Merging | Evo-ViT | 79.4(−0.4) | 2.9(−1.7) |
| | | MG-ViT | 80.8(+1.0) | 2.4(−2.2) |

**Experiment on images with higher resolution.** We conduct experiments on higher resolution images, as shown in Table 13, demonstrating the continued effectiveness of our proposed multi-granularity strategy on higher resolution images.

## 7.5 Computational Complexity Analysis

In this discussion, we disregard the patch embedding block. The computational complexity of a transformer layer responsible for processing tokens can be calculated as follows:

$$\Omega(SA(tokens)) = 4NC^2 + 2N^2C,$$
$$\Omega(FFN(tokens)) = 8NC^2. \tag{18}$$

Based on provided equations, it can be observed that increasing the dimension $C$ leads to a quadratic growth in computational complexity. Therefore, if the dimension $C$ is expanded to $2\times$, the computational cost will approximately increase by $4\times$, as discussed in the feature reuse part.

In the PPSM of MG-ViT, we introduce the three-way decision slimming to reduce the number of tokens. Specifically, based ratios $r_{pos}$ and $r_{neg}$, all tokens are divided into positive, negative, and boundary tokens. Positive tokens remain unchanged. Boundary tokens are merged to the half of their original quantity following the guidance of the token match-merge scheme. Negative tokens are merged into a single token, and their quantity can be neglected. Therefore, the computational complexity in the transformer layer following the first PPSM can be described as follows:

$$N' = r_{pos}N + \frac{1 - r_{pos} - r_{neg}}{2}N$$
$$\Omega(PPSM^{1st}) = 12N'C^2 + 2N'^2C. \tag{19}$$

We employ DeiT-S as the backbone and insert PPSM before the self-attention (SA) layers in the 4th, 7th, and 10th layers. The positive, boundary, and negative tokens are divided according to a ratio of 5:4:1. The computational complexity of DeiT-S and our MG-ViT are:

$$\Omega(DeiT) = 144NC^2 + 24N^2C,$$
$$\Omega(STViT) = 91.2NC^2 + 11.1N^2C. \tag{20}$$

## 7.6 Discussion of Ratio $r$

$r_h$ **and** $r_t$. Each head patch is subsplit into 9 fine patches, while each middle patch is subsplit into 4 medium patches. The majority of important information is concentrated in a few head patches, partially distributed among middle patches, and only a small portion exists in tail patches. Consequently, we mandate that the count of middle patches exceeds half of the total number. Additionally, as our aim is to develop an efficient ViT framework, we strive to minimize the total number of patches obtained after subsplitting compared to the 196 tokens (the official splitting method $14\times14$). Considering these factors, we set $r_h < 0.2$. Furthermore, based on the results presented in Fig.1(a) and Fig.1(c), we set $0.3 < r_h < 0.5$. We conducted experiments on the origin MG-ViT (without PPSM) and selected several competitive control group results to showcase in table 15.

Table 15: Accuracy and FLOPs with different values of $r_h$ and $r_t$.

| $r_h \backslash r_t$ | 0.05\0.3 | 0.05\0.4 | 0.1\0.3 | 0.1\0.4 | 0.15\0.35 |
|---|---|---|---|---|---|
| Acc.(%) | 80.7 | 80.5 | 81.0 | 81.0 | 81.1 |
| FLOPs(G) | 3.6 | 3.5 | 4.1 | 3.9 | 4.2 |

$r_{pos}$ **and** $r_{neg}$. According to the discussion in Section 6.3, we exclude the match-merge of boundary tokens and only consider the merging of negative tokens. When $r_{neg} = 0.2$, the number of tokens in the last layer of MG-ViT decreases to 51.2% of the original count. Therefore, the quantity of negative tokens should be relatively small, and we set $r_{neg} < 0.2$. On the other hand, if $r_{pos} = 0.5$, the minimum number of tokens in the last layer of MG-ViT should be 12.5% of the original count, which serves as the lowest limit for normal operation of MG-ViT. As a result, we set $0.5 \leq r_{pos} \leq 0.7$. Our experimental findings, as presented in Table 16, include several competitive control group results.

Taking all aspects into consideration, we set $r_h = 0.1$, $r_t = 0.4$, $r_{pos} = 0.5$ and $r_{neg} = 0.1$.

## 7.7 Three-Way Decision

Three-way decision is a decision mechanism in rough set theory. Let $U$ be the set of all tokens, and given a pair of thresholds $r_{pos}$ and $r_{neg}$. Let $e(x)$ denote the rank of token $x$ based on CASR, and

Table 16: Accuracy and FLOPs with different values of $r_{pos}$ and $r_{neg}$.

| $r_{pos} \backslash r_{neg}$ | 0.5\0.1 | 0.5\0.15 | 0.6\0.05 | 0.6\0.1 | 0.7\0.05 |
|---|---|---|---|---|---|
| Acc.(%) | 80.8 | 80.6 | 81.0 | 81.0 | 81.1 |
| FLOPs(G) | 2.4 | 2.3 | 2.9 | 2.8 | 3.2 |

$num$ represent the total number of tokens in the PPSM. We can describe the positive, negative, and boundary domains as follows:

$$Pos = \{x \in U \mid e(x) \leq num \cdot r_{pos}\},$$
$$Neg = \{x \in U \mid e(x) \geq num \cdot (1 - r_{neg})\}, \quad (21)$$
$$Bdy = \{x \in U \mid num \cdot r_{pos} < e(x) < num \cdot (1 - r_{neg})\}.$$

For tokens in the positive and negative domains, we employ deterministic operations to either maintain them unchanged or merge them into one token. This is because we have determined their redundancy status. For tokens in the boundary domain whose redundancy status has not been determined, we employ undeterministic operation, token match-merge. Token match-merge strikes a balance between maintaining and merging, representing a form of slow merge that only merges similar tokens. If the newly generated boundary tokens are subsequently divided into the positive and negative domains in the next PPSM, they will undergo deterministic operations.

