# OpenReview forum: "MG-ViT: A Multi-Granularity Method for Compact and Efficient Vision Transformers"
_NeurIPS.cc/2023/Conference — NeurIPS 2023 poster_

### Official Review · Reviewer_L7yG · 2023-06-27

**Soundness:** 3 good
**Presentation:** 3 good
**Contribution:** 2 fair
**Rating:** 5
**Confidence:** 4

**Summary:**

This paper improves the inference efficiency of Vision Transformers by token pruning/merging. Specifically, the authors proposed a Multi-Granularity method, which splits an image into patches of different sizes and merges similar tokens. Extensive experiments and ablation studies on ImageNet have demonstrated the advantage of their method.

**Strengths:**

1. Simple idea but works. The proposed method achieves better speed-accuracy trade-off than many existing methods.
2. The experiments are comprehensive, especially for the ablation study.
3. This paper is easy to follow. The overall presentation is of good quality.

**Weaknesses:**

1. Despite the great performance, the evaluation is only conducted on one scale, i.e., the small version of DeiT and LV-ViT. It is not clear if the proposed method can generalize to other scales, such as DeiT-Ti and DeiT-B.
2. Lack throughput comparison in Table 3. FLOPs may not reflect the real speed on hardware. It is important to directly evaluate the inference speed when comparing with other related works.
3. Due to token merging, the proposed method may not be general for downstream dense prediction tasks.

**Questions:**

1. I would like to know the throughput/model parameter comparison in Table 3.
2. Unlike this paper which trains a model from scratch with 300 epochs, DynamicViT utilises pretrained weights to finetune their model. How the performance would be if the MG-ViT is also finetuned based on pretrained DeiTs.

**Limitations:**

Yes

---

> ### Author Rebuttal · Authors · 2023-08-10
>
> We greatly appreciate your meticulous review and professional suggestions! We address the questions and clarify the issues accordingly as described below.
>
> **Weakness 1**: Your suggestion is correct. In order to demonstrate the generality of the method, we indeed need to evaluate it on backbones of other scales. Therefore, we have supplemented the experimental evaluations of the remaining sizes of DeiT and LV-ViT. A total of 4 backbones are covered, with experiments completed for DeiT-Ti, DeiT-B, and LV-ViT-M (please refer to Table 13 in the PDF). Due to limitations in computational resources and time, we have temporarily not conducted experiments for LV-ViT-L; we will add its experimental results in the revised version. Thanks!
>
> **Weakness 2 & Question 1**: We apologize for not fully presenting our experimental results. Based on your suggestion, we have added a comparison of throughput and parameters in Table 3. (Please refer to Table 16 in the PDF.) Thank you!
>
> **Weakness 3**: Your perspective is correct. The proposed framework is not suited for handling dense tasks such as detection and segmentation. Since we focus on compressing ViT to improve efficiency in this work, rather than proposing a new general ViT structure, similarly to classical papers[r1-r7] in ViT compression, which mainly evaluated classification tasks. Therefore many designs are inclined towards classification. For instance, our proposed PPSM, which further reduces the computational overhead of ViT, is based on classification task. In our subsequent research, we are actively exploring the extension of the multi-granularity strategy to dense downstream tasks. However, for such scenarios, PPSM may not be applicable. Fortunately, Reviewer cCy5 provided a valuable reference 4, which effectively demonstrates that token pruning from ViT can be applied to dense tasks such as detection and segmentation. This leads me to Continue infer whether merging could also be explored in dense tasks? Although we have some general ideas about extending multi-granularity strategy to dense downstream tasks, they are not fully matured, and there are numerous details that need to be addressed. I believe this will be a highly challenging yet meaningful work.
>
> **Question 2**: We greatly appreciate your attention to details, as you helped us discover a slip in the paper. Similar to DynamicViT, MG-ViT is also achieved through fine-tuning. Therefore, the results we presented are obtained after MG-ViT fine-tuning on pre-trained DeiT or LV-ViT. We will thoroughly find similar slips and further polish our presentation. A well-written paper will be presented in the revised version. Thank you once again!

---

> > ### Author Response · Authors · 2023-08-21
> >
> > once again, we would appreciate your review. In your feedback, you suggested us to append the experimental results for other scales of DeiT and LV-ViT. Due to limitations in computational resources and time, we were unable to complete the experiment for LV-ViT-L during the rebuttal phase. However, we have now completed it. The results are as follows:
> >
> > |Method | $\quad$Threshold($\epsilon$)$\quad$ | Accuracy(%)$\quad$  | FLOPs(G) | Throughput(img./s) |
> > | - | :-: | :-: | :-: | :-: |
> > | LV-ViT-L |—| 85.3 | 58.8 | 189|
> > | MG-ViT |0.56 | 85.3(+0.0) | 30.1(-49%) |359(+1.91×) |
> > | MG-ViT |0.67| 85.9(+0.0) | 36.5(-38%) |305(+1.61×) |
> > | MG-ViT |1 | 86.3(+1.0) | 47.6(-19%) | 263(+1.39×)|
> >
> > It can be observed that our proposed framework remains applicable to LV-ViT-L. (288×288)
> >
> > We are looking forward to further discussions with you in these final moments, which will significantly contribute to supporting our work. Thank you!
> >
> > Best wishes,
> >
> > All authors of Paper 6032.

---

> > > ### Comment · Reviewer_L7yG · 2023-08-22
> > > **Official Comment by Reviewer L7yG**
> > >
> > > Thank you for the rebuttal and the additional results. I am happy to see that the authors addressed most of my concerns in the rebuttal.
> > >
> > > However, according to the response, the models are fine-tuned based on pre-trained weights. In lines 265-266, the authors initially claimed that "all models are trained from scratch." From my point of view, this is a significant difference that will affect my rating of this paper.
> > >
> > > In short, training from scratch while achieving this good speed-accuracy trade-off would be impressive. Fine-tuning with a few epochs is also acceptable, as it is a common practice and can help the model to ensure performance. However, 200 or even more epochs of training (as reported in the supplements) is an expensive training cost. Therefore, the comparison with other related works does not seem fair, even if it achieves better performance.
> > >
> > > As shown in the general response PDF, under the similar FLOPs, MG-ViT achieves similar throughput. It is not clear whether this gain is obtained from more training iterations or the advantage of the proposed method.
> > >
> > > Apart from this issue, I cannot totally agree with Reviewer V3HF but also have the same concern about the significance of this paper. At this moment, only experimenting on ImageNet classification seems minor (a lot of similar works with good performance). It would be greatly appreciated if the authors could find more clear advantages of their method in other tasks.
> > >
> > > Considering the above, I will keep my initial rating.

---

### Official Review · Reviewer_cCy5 · 2023-07-05

**Soundness:** 3 good
**Presentation:** 1 poor
**Contribution:** 2 fair
**Rating:** 5
**Confidence:** 5

**Summary:**

This paper presents a multi-granularity method to develop efficient vision Transformers. A two-stage inference pipeline is designed to handle easy and difficult samples in different ways. The method is evaluated on ImageNet based on DeiT and LVViT.

**Strengths:**

- The motivation presented in the Introduction looks convincing and insightful.

- The method achieves good performance on ImageNet classification and can outperform previous methods.

**Weaknesses:**

- The writing of this paper can be further improved. There are quite a few grammar errors and typos. Many sentences are confusing or hard to understand. We list some of them: line 65, "we propose MG-ViT 2"; line 80, "Simply put, based on CASR, tokens are divided ...".

- Although the method is well-motivated, the proposed method is not very new: there are several existing methods to use classification probability to estimate the confidence of a prediction and use it to adjust computation [r1]; the token merging and multi-scale inference framework have also been explored in previous work like DVT and [r2].

- The experimental results are not very impressive and competitive in 2023 since recent efficient vision Transformers are usually compatible with more diverse architectures like Swin/ConvNeXt [r1,r3] and more complex tasks like detection and segmentation [r3, r4]. The claim in line 87 "MG-ViT is a dynamic and versatile framework that can be applied to most ViT models" is also not well supported since it seems non-trivial to apply the method to hierarchical Transformers.

[r1] Token Merging: Your ViT But Faster, ICLR 2023.

[r2] MIA-Former: Efficient and Robust Vision Transformers via Multi-Grained Input Adaptation, AAAI 2022.

[r3] Dynamic Spatial Sparsification for Efficient Vision Transformers and Convolutional Neural Networks, TPAMI.

[r4] Revisiting Token Pruning for Object Detection and Instance Segmentation, arXiv:2306.07050

**Questions:**

Please refer to my comments above.

**Limitations:**

The limitations of the proposed method have not been discussed.

---

> ### Author Rebuttal · Authors · 2023-08-09
>
> Thank you very much for your careful and insightful review! I greatly benefited from your feedback, especially the reference 4 you provided. We address the questions and clarify the issues accordingly as described below.
>
> **Weakness 1**: Thanks very much for your careful review! We apologize for the confusion caused by our presentation during your review. For instance, your mention of line 65, "we propose MG-ViT 2," where 2 refers to the reference of Figure 2 set in our LaTeX document. We will thoroughly review similar errors and further polish our presentation. A well-written paper will be presented in the revised version. Thank you once again!
>
> **Weakness 2**: ① We agree that the idea of using classification probability to estimate the confidence of a prediction and adjusting computation is not new. However, in our case, we chose to compress the original non-hierarchical ViT for classification tasks. This inherently includes the class token, as it integrates information from each token for classification. Consequently, employing classification probability (class attention score) is the sole method to achieve a balance between performance and computation. ② Our work is inspired by 3 observations during classification. We combined the idea of multi-granularity with common techniques such as multi-stage and token merging to achieve the classification task while skillfully balancing computational costs and performance. It's a simple but effective approach. This exploration has offered us a new perspective on problem-solving. ③ While DVT and MIA-Former have explored multi-scale inference framework, we beg to differ that our proposed multi-granularity strategy aligns with them. DVT and MIA-Former integrate multi-scale tokens between different transformer layers, whereas, we integrate multi-granularity patches in the same transformer layer, which is a new attempt. Therefore, we believe our work is novel.
>
> **Weakness 3**: ① We have found several works [r9-r22] published in top journals and conferences in 2023, including 2 recently submitted articles [r18, r21] on arXiv this month. These works have employed various backbones such as ViT, CNN, Swin-T, GNN and ConvNeXt. Upon comparison (Please refer to Table 17 in PDF) with their performance, we find that MG-ViT remains competitive. Furthermore, our work focuses on compressing ViT, and the methods we compare with in our paper, such as eTPS [r23] and STViT[r24], were also published in 2023. Hence, within the domain of ViT compression, our method currently achieves state-of-the-art outcomes. ② Since we focus on compressing ViT to improve efficiency in this work, rather than proposing a new general ViT structure, similar to classical papers [r1-r7] in ViT compression, which mainly evaluated classification tasks, we've adopted the same approach. However, we are also actively attempting to extend the multi-granularity strategy to downstream tasks such as detection and segmentation. This topic is also being explored in our subsequent research. Just like the reference 3 you provided, which is the subsequent exploration of DynamicViT. It must be acknowledged that this general idea still requires many details to be worked out. However, we sincerely appreciate your provision of Reference 4, which has given me an entirely new perspective. The work in Reference 4 has effectively demonstrated that token pruning from ViT can be applied to dense tasks such as detection and segmentation. I will delve further into this work and attempt to integrate it into my future research. Once again, sincere gratitude for your provision! ③ Our proposed model can be combined with hierarchical ViT such as Swin-T. For the detailed approach and analysis, please refer to the response of Weakness 1 from Reviewer V3HF. However, we find that combining with hierarchical ViT is unnecessary for classification. The merging in hierarchical ViT considerably diminishes the advantages of multi-granularity in feature representation.

---

> > ### Comment · Reviewer_cCy5 · 2023-08-20
> >
> > Thanks for the detailed feedback. It is nice to see further discussions about the throughput of the proposed models and the compatibility of hierarchical models. Although the proposed method is not that new, I think the presented contribution is solid. I would like to upgrade my rating to Borderline Accept.

---

> > > ### Author Response · Authors · 2023-08-20
> > >
> > > We sincerely value your constructive feedback and meticulous review, and we extend our heartfelt appreciation for further support by upgrading the rating and recognizing our contribution. We hope that this paper achieves a satisfactory result, not in vain of your support and efforts. Thank you!
> > >
> > > Best wishes,
> > >
> > > All authors of Paper 6032.

---

### Official Review · Reviewer_V3HF · 2023-07-08

**Soundness:** 3 good
**Presentation:** 3 good
**Contribution:** 3 good
**Rating:** 4
**Confidence:** 5

**Summary:**

The proposed MG-ViT framework introduces a two-stage multi-granularity approach for Vision Transformers (ViTs), aiming to balance performance and computational cost. Initially, an image is divided into patches for basic classification. If the image isn't accurately recognized, the system initiates a more detailed patch analysis stage.

To reduce the computational cost, a three-way decision slimming method is proposed. A token match-merge scheme also aids in making the ViT more compact and efficient. Tests on ImageNet show that without performance loss, MG-ViT reduces the FLOPs of LV-ViT by 47% and DeiT by 56%, demonstrating significant computational cost reduction.

**Strengths:**


-The underlying motivation for the study is lucid, and the proposed method displays novelty.

-The manuscript is well-written and easy to comprehend.

-The experimental results highlight an impressive balance between accuracy and speed offered by the proposed method

**Weaknesses:**

While the Multi-Granularity strategy is well-intentioned and produces promising classification results, its application to other vision downstream tasks such as segmentation presents significant challenges.

Additionally, there is a lack of results regarding window-based Vision Transformers like the Swin Transformer. This could be due to the complexity and difficulty in applying this strategy to such models.

**Questions:**

Please see weakness.

**Limitations:**

Please see weakness.

---

> ### Author Rebuttal · Authors · 2023-08-10
>
> We would like to thank you for taking the time to review our work. We address the questions and clarify the issues accordingly as described below.
>
> **Weakness1：combining our proposed framework and Swin-T.**
>
> We have previously investigated the integration of our proposed framework with Swin-T and found it to be feasible. In a general sense, we have made the following modifications to the framework:
>
> 1. Replaced the original ViT in MGIS with a window-based hierarchical ViT, namely Swin-T.
> 2. Diverging from the 7×7 splitting scheme discussed in the paper for SGIS, we employed an 8×8 scheme. The patches resulting from the 8×8 splitting are referred to as coarse patches.
> 3. In MGIS, contiguous 2×2 coarse patches constitute a window.
> 4. Utilizing CSAR, we selected the important coarse patches and further subsplitted each of these important coarse patches into four 4 patches through a 2×2 subsplitting procedure.
> 5. During patch merging in Swin-T, each coarse patch (token) was replicated 4× and treated as fine patches (tokens) to merge.
>
> We evaluated the model performance on the ImageNet. It was observed that the LV-ViT-S-based framework (referred to as MG-ViT) proposed in the paper achieves a reduction of 47% in FLOPs without sacrificing performance. In contrast, the Swin Transformer-S-based framework only achieves a 24% reduction in FLOPs and also lags behind MG-ViT in terms of accuracy. We attribute this discrepancy to two main factors:
>
> Firstly, the interaction between coarse and fine-grained patches (tokens) can only occur at shallower layers. With the continuous merging of patches, the granularity differences between tokens diminish, significantly eroding the benefits of multi-granularity in feature representation. However, the proposed PPSM effectively mitigates this issue by allowing patches of varying granularity to interact over several transformer layers before merging.
>
> Secondly, this discrepancy can also be attributed to inherent characteristics of Swin-T itself. As has been lamented by some researchers, the windowed attention mechanism deviates from the original intent of the ViT design, transforming global interactions into local interactions. In contrast, employing the original ViT circumvents such an issue from arising.
>
> Our work in the paper offers fresh insights, challenging the notion that hierarchical ViT is superior. The original non-hierarchical ViT coupling with the multi-granularity method can achieve better results, simple but work. In object detection, ViTDet[r8] proposed by Kaiming He also demonstrates that hierarchical structures like Swin-T are not the only efficient ways to accomplish visual tasks.
>
> **Weakness2**：While incorporating a multi-granularity strategy into downstream tasks presents significant challenges, it is not insurmountable. As previously discussed, the integration of our proposed framework with Swin-T offers a promising foundation for implementing the multi-granularity strategy in downstream tasks. Moreover, we extend our gratitude to Reviewer cCy5 for furnishing a valuable reference 4, which effectively illustrates the applicability of token pruning from ViT to dense tasks like detection and segmentation. This contribution has provided valuable insights and bolstered our confidence. Consequently, we eagerly anticipate observing the performance of the multi-granularity strategy in diverse downstream tasks. However, this topic will be explored in our subsequent research. Since we focus on compressing ViT to improve efficiency in this work, rather than proposing a new general ViT structure, similar to classical papers[r1-r7] in ViT compression, which mainly evaluated classification tasks, we've adopted the same approach.

---

> > ### Comment · Reviewer_V3HF · 2023-08-17
> >
> > I really appreciate the rebuttal of authors. But the paper in the current stage needs a major revision for providing results of pixel level donwstream tasks. Therefore, I keep my original rating and lean towards rejection.

---

> > > ### Author Response · Authors · 2023-08-19
> > >
> > > Thanks for your response. On one hand, this work focuses on compressing the plain ViT rather than proposing a new general ViT structure. Classic compression works on plain ViT are mainy evaluated on classification tasks. On the other hand, as our motivation originates from observing phenomena on the classification of plain ViT, our design aligns with the structural characteristics of plain ViT. Therefore, our proposed compression framework MG-ViT based on plain ViT cannot inherently extend to downstream tasks such as detection and segmentation without any modification.
> > >
> > > To sum up, that is the reason why we are unable to provide experimental results of MG-ViT on downstream tasks. In our subsequent research, we are actively exploring in the transfer of the multi-granularity strategy to detection and segmentation tasks, we have discussed its feasibility in our responses to you and other reviewers. One work cannot address all issues; our focus lies in utilizing the multi-granularity strategy to compress plain ViT in order to achieve better classification performance with lower computational costs.
> > >
> > > We are trying our best to address your concerns and looking forward to further discussion. Thanks!
> > >
> > > Best wishes,
> > >
> > > All authors of Paper 6032.

---

> > > > ### Comment · Reviewer_V3HF · 2023-08-20
> > > >
> > > > First, plain ViT can be easily extended to downstream tasks.
> > > >
> > > > Second, many papers e.g., FasterViT mentioned by qw7Z all conduct experiments on downstrem tasks. I don't think a backbone paper reaches the bar of Neurips if only classification results are provided.

---

> > > > > ### Author Response · Authors · 2023-08-20
> > > > >
> > > > > We extend our sincere appreciation for your response! We will value your suggestions and conduct a further exploration in this aspect.
> > > > >
> > > > > Best wishes,
> > > > >
> > > > > All authors of Paper 6032.

---

### Official Review · Reviewer_J3bo · 2023-07-11

**Soundness:** 3 good
**Presentation:** 3 good
**Contribution:** 2 fair
**Rating:** 5
**Confidence:** 4

**Summary:**

This paper presents MG-ViT, a way to save computational cost without affecting the performance of ViT-based methods. MG-ViT is a two-stage model, the first stage is a lightweight ViT for single-granularity inference, and the second stage network for multi-granularity inference.

**Strengths:**

1、A series of experiments are conducted to demonstrate the influence of tokens of different IIC and granularity on network performance, which provides a clear motivation for the proposed MG-ViT.

2、The design of MGIS, including multi-granularity patch subsplitting, PPSM, and token match-merge scheme is natural and reasonable.

3、The ablation experiments of the proposed components are sufficient.


**Weaknesses:**

1、In recent years, a series of ViT-based methods based on window attention (Such as Swin Transformer, and Slide-Transformer[1]) have been proposed, which have better classification performance and lower computational cost than the original ViT (DeiT). This paper can better prove the effectiveness of MG-ViT, if it can be compatible with the frameworks based on window attention especially the current sota methods (Slide-Transformer).

2、This paper can better prove the effectiveness of MG-ViT, if it can be compatible with ViTs pretrained by massive data (JFT-3B or LAION) or self-supervised pre-trained (Such as MAE[2], DINOv2), which achieves better performance (more than 85% acc on ImageNet).


[1] Pan X, Ye T, Xia Z, et al. Slide-Transformer: Hierarchical Vision Transformer with Local Self-Attention[C]//Proceedings of the IEEE/CVF Conference on Computer Vision and Pattern Recognition. 2023: 2082-2091.

[2] He K, Chen X, Xie S, et al. Masked autoencoders are scalable vision learners[C]//Proceedings of the IEEE/CVF conference on computer vision and pattern recognition. 2022: 16000-16009.

**Questions:**

The MG-VIT proposed in this paper is novel enough, but the effectiveness of the proposed method needs to be validated under stronger baselines.

**Limitations:**

Refer to weakness

---

> ### Author Rebuttal · Authors · 2023-08-09
>
> Thank you for your review, recognition, and suggestions. We address the questions and clarify the issues accordingly as described below.
>
> **Weakness 1**: ① It is evident that hierarchical ViT models based on window attention, such as Swin-T, surpass the original ViT in many tasks or performance aspects. We have also explored combining our multi-granularity strategy with Swin-T. For the detailed approach and analysis, please refer to the response of Weakness1 from Reviewer V3HF. However, we find that combining with Swin-T is unnecessary for classification. The merging in Swin-T considerably diminishes the advantages of multi-granularity in feature representation. ② slide-transformer [r22] proposes a novel local attention module, Slide Attention. We have appended experimental results comparing the performance of MG-ViT and Slide-Transformer on classification tasks (Please refer to Table 17 in PDF). In the future, we are willing to carefully consider your suggestion to make the multi-granularity strategy and Slide-Transformer compatible. Thank you!
>
> **Weakness 2**：Indeed, utilizing more massive datasets like JFT-3B for pretraining can improve the performance. Currently, models pre-trained using datasets like Laion, such as OpenCLIP and the aforementioned DINOv2, have achieved encouraging outcomes in the ImageNet classification task. There are certain aspects to learn from them, but we haven't yet specifically considered how to combine them. We will carefully contemplate this in the future. Article [r25] found that after employing appropriate MAE pre-training and fine-tuning on ImageNet, the performance of the original ViT-Tiny surpassed that of various lightweight CNNs and meticulously designed ViT variants.
> In line with your suggestion, we will consider selecting appropriate MAE pre-training in our next work. Thank you!
>
> **Question 1**: Thank you for recognizing the novelty of our work. Following your and Reviewer cCy5's common suggestion, we have found some recent works published in 2023 as stronger baselines and appended relevant experiments (Please refer to Table 17 in the PDF). Furthermore, our work focuses on compressing ViT, and the methods we compare with in our paper, such as eTPS [r23] and STViT [r24], were also published in 2023. Hence, within the domain of ViT compression, our method currently achieves state-of-the-art outcomes.

---

> > ### Comment · Reviewer_J3bo · 2023-08-20
> >
> > Part of my concerns have been sufficiently addressed by the author's rebuttal.
> >
> > However, since the pre-trained ViT does not change the structure of the ViT, I think the combination of MG-ViT and a stronger pre-trained ViT is experimentally feasible.

---

> > > ### Author Response · Authors · 2023-08-20
> > >
> > > We would like to thank you for your response. During the rebuttal stage, the reason we didn't append experiments was that we couldn't obtain the MAE pre-trained models for DeiT-S and LV-ViT-S in a short period of time. Through inference and findings from existing works, it is evident that utilizing MAE pre-trained models can indeed prove the performance of MG-ViT. However, such an approach cannot provide a fair comparison under the same backbone.
> > >
> > > Nevertheless, in order to address your concern better, we are conducting supplementary experiments using MAE-Tiny[r25], the MAE pre-trained models for ViT-Tiny, which will confirm your idea. We expect to release the experiment results in about eight hours.
> > >
> > > Please stay tuned! We also eagerly look forward to your response and further discussion. thank you!

---

> > > > ### Comment · Reviewer_J3bo · 2023-08-20
> > > >
> > > > There are many pre-trained ViT-based mae models in HuggingFace ( https://huggingface.co/facebook/vit-mae-huge) that achieve high classification performance on ImageNet. I think the ViT structure of these maes is not fundamentally different from DeiT-S, and they can be used to combine with MG-ViT.

---

> > > > > ### Author Response · Authors · 2023-08-20
> > > > >
> > > > > We apologize for keeping you waiting for a long time. We sincerely appreciate your practical suggestions. Regrettably, we have been unable to obtain the MAE pre-trained models for DeiT-S and LV-ViT-S in communities like Hugging Face, GitHub and others. As a result, we utilized the available MAE-Tiny model (a MAE pre-trained model for DeiT-Tiny, which is accessible from the resource repository mentioned in Reference 25[r25]) in combination with our proposed MG-ViT framework. The experimental results are as follows:
> > > > >
> > > > > | Method| Threshold($\epsilon$)$\quad$ | Accuracy(%)$\quad$  | FLOPs(G) | Throughput(img./s) |
> > > > > | - | :-: | :-: | :-: | :-: |
> > > > > | MAE-Tiny |—| 78.0 | 1.3 | 3693|
> > > > > | MG-ViT(MAE-Tiny)$\quad$ |0.54 | 78.0(+0.0) | 0.68(-48%) |6510(+1.76×) |
> > > > > | MG-ViT(MAE-Tiny) |1 | 78.8(+0.8) | 1.26(-3%) | 4057(+1.10×)|
> > > > >
> > > > > We found that after combining the MAE-Tiny model with our multi-granularity framework, MG-ViT(MAE-Tiny) has better performance compared to MG-ViT(DeiT-S) (Please refer to Table 13 in the PDF). During the experiment, we found that utilizing a stronger pre-trained model contributes significantly to throughput enhancement without any loss of accuracy. With the enhanced recognition capability of ViT in the SGIS, there is more leeway for lowering thresholds without sacrificing accuracy. Additionally, This leads to more images completing inference in SGIS, thus saving computational costs and accelerating inference speed. This finding further substantiates your viewpoint that "the effectiveness of MG-ViT can be better proven by combining it with a stronger pre-trained ViT." We conducted experiments solely on the smallest MAE pre-trained model; we infer that this phenomenon would be even more obvious when applied to larger MAE pre-trained models.
> > > > >
> > > > > As the discussion deadline is less than 24 hours away, additionally, we are unable to access the MAE pre-trained models for other scale of DeiT and LV-ViT, we regretfully cannot provide more experimental results. However, we have the same perspective as you that "the ViT structure of these MAEs is not fundamentally different from DeiT-S." If this is a matter of considerable interest and expectation for you, we commit to extending the length in the final version to show the performance of MG-ViT combined with MAE-ViT-B, MAE-ViT-L, DINO-ViT-S, DINO-ViT-B, DINO2-ViT-B, and DINO2-ViT-L.
> > > > >
> > > > > We sincerely appreciate the guidance and suggestions you've provided. We are looking forward to further discussions with you in these final moments, which will significantly contribute to improving our work. Thank you!
> > > > >
> > > > > Best wishes,
> > > > >
> > > > > All authors of Paper 6032.

---

> > > > > > ### Comment · Reviewer_J3bo · 2023-08-21
> > > > > >
> > > > > > Thanks for your reply. Based on the results so far, I'll keep my rating at borderline accept. I am looking forward to seeing more results about combining MG-ViT with large maes in the final version.

---

> > > > > > > ### Author Response · Authors · 2023-08-21
> > > > > > >
> > > > > > > We would like to sincerely appreciate your constructive suggestions, which have illuminated us more avenues for enhancing model performance and greatly benefited us. We hope that this paper achieves a satisfactory result, not in vain of your expectations and suggestions. Thank you!
> > > > > > >
> > > > > > > Best wishes,
> > > > > > >
> > > > > > > All authors of Paper 6032.

---

> ### Comment · Area_Chair_eoA6 · 2023-08-20
> **Please respond to the content of the authors‘ response.**
>
> The discussion period is approaching. We expect that all such comments receive a reply within the author discussion period that ends on Monday. Thank you so much for your great effort made to NeurIPS 2024.

---

### Official Review · Reviewer_qw7Z · 2023-07-16

**Soundness:** 3 good
**Presentation:** 3 good
**Contribution:** 3 good
**Rating:** 6
**Confidence:** 5

**Summary:**

This paper introduced a new vision transformer model, denoted as MG-ViT, which consists of a two-stage approach to balance the trade-off between the computational cost and performance. Specifically, it proposes a two-stage approach, in which a single-granularity inference stage is used to determine if the image can be correctly classified. If not, then a second multi-granularity stage is employed. The model is trained and tested on ImageNet-1K dataset and reportedly achieve competitive performance.

**Strengths:**

1. The paper is well-written and easy to follow.

2. The idea of improving vision transformer efficiency is important and has merits.

3. The authors present comprehensive ablations studies and analysis of their proposed model.

**Weaknesses:**

1. The proposed effort lacks novelty as it essentially proposes a two-stage approach with different level of granularity to strike a balance between accuracy vs efficiency. In the proposed framework, there are several assumptions that may limit the applicability of this work. First, it is not clear how to determine the confidence judgement threshold in the first stage which can significantly impact the performance. Second, the head.middle and tail patchification scheme cannot be generalized to all use-cases and can indeed introduce undesired biases that can negatively impact the model's performance.

2. One of the main arguments of this work in justifying its usefulness is the efficiency aspect. However, the authors only report the gains in terms of number of FLOPs which does not include the entire picture. In fact, the throughput comparison is only shown in Figure 10 of the supplementary materials and without any actual quantitative comparisons. Since the method is comprised of two stages, it is expected that the throughput is negatively impacted.

3. The method is only evaluated for smaller model regime and it is not clear whether it scales to larger models (e.g. ViT-L, ViT-H, ViT-G).

4. The experimental results are not convincing enough and don't validate the effectiveness of this work.

**Questions:**

1. The proposed model is only evaluated for classification. Can the same ideas be extended to down-stream tasks such as detection and segmentation ?

2. Typical ImageNet images with resolution 224 x 224 are used for training and evaluation of the proposed model. How about higher resolution images ?

3. Can the proposed model be used with hierarchical vision transformers such as Swin Transformers ?

**Limitations:**

Yes

---

> ### Author Rebuttal · Authors · 2023-08-09
>
> We would like to thank you for taking the time to review our work. We address the questions and clarify the issues accordingly as described below.
>
> **Question 1**: Transferring the multi-granularity strategy into dense tasks might be considered feasible. However, this topic will be explored in our subsequent research. Since we focus on compressing ViT to improve efficiency in this work, rather than proposing a new general ViT structure, similar to classical papers[r1-r7] in ViT compression, which mainly evaluated classification tasks, we've adopted the same approach. we're also eager to observe the performance of the multi-granularity strategy in downstream tasks, making it exciting and interesting for future work.
>
> **Question 2**: Thanks for your suggestion. The average resolution of images in ImageNet is 469×387. Standard works[r1-r7] on ViT compression all have employed a resolution of 224×224 for training and evaluation. However, a few studies have also fine-tuned and evaluated at 384 resolution. As a result, we have appended the result at 384 resolution. Please refer to Table 15 in the PDF.
>
> **Question 3**: Our proposed model can be combined with hierarchical ViT such as Swin-T. For the detailed approach and analysis, please refer to the response of Weakness 1 from Reviewer V3HF. However, we find that combining with hierarchical ViT is unnecessary for classification. The merging in hierarchical ViT considerably diminishes the advantages of multi-granularity in feature representation. This work offers fresh insights, challenging the notion that hierarchical ViT is superior. The original non-hierarchical ViT coupling with the multi-granularity method can achieve better results, simple but work. In object detection, ViTDet[r8] proposed by Kaiming He also demonstrates that hierarchical structures like Swin-T are not the only efficient ways to accomplish visual tasks.
>
> **Weakness 1**: ① While various variants of ViT have emerged, our motivation differs from theirs. Inspired by 3 observations during classification, we combined the idea of multi-granularity with common techniques like multi-stage and token merging, accomplishing classification tasks while adeptly balancing the computational cost and performance. Simple but work. This exploration provided us with a new view of problem-solving. And we made the first attempt to integrate multi-granularity patches in the same transformer block. Therefore, we believe our work is novel, gratefully, which is also appreciated by Reviewer J3bo and V3HF. ② In Table 2, we show the impact of thresholds on accuracy, FLOPs, and throughput. Taking MG-ViT with a DeiT-S backbone as an example, when $\epsilon=0.49$, MG-ViT reduces FLOPs by 56% and increases throughput by 1.79x without any accuracy loss. We regret the absence of a unified method for determining the threshold as a hyperparameter. However, we provide additional data (Table 14 in PDF) to assist in threshold determination based on hardware configuration and other specific requirements. ③ During the inference stage, the patchification scheme (in MGIS) is not applied to all validation use-cases; it is only used for "complex" cases. Figure 8 provides the visualization of both “complex” and “simple” cases. This is because if an image can be well recognized in SGIS, there is no need to execute MGIS for the head, middle, tail patch identification and subsplitting. This plays a pivotal role in the effective balance of performance and computation achieved by MG-ViT. Additionally, during the training stage, as claimed in line 253, "During the training of MG-ViT, setting $\epsilon=1$ makes MGIS be executed for every input image." all training use-cases (images) are employed in the training stage (especially in MGIS). Therefore, MG-ViT avoids introducing undesired biases during training, fairness can be ensured.
>
> **Weakness 2**: In Table 2, we quantitatively show the throughput (TP) of MG-ViT. It can be observed that our strategy enhances throughput, and the two-stage method does not negatively affect it throughput. Responding to your and Reviewer L7yG's feedback, we've incorporated throughput and parameter comparisons in Table 3. (Please refer to Table 16 in PDF)
>
> **Weakness 3**: Classical works [r1-r7] on ViT compression have chosen DeiT, LV-ViT, and T2T (later replaced by LV-ViT) as backbones to develop efficient ViTs. Therefore, we have supplemented the experimental evaluations of the remaining sizes of DeiT and LV-ViT. A total of 4 backbones are covered, with experiments completed for DeiT-Ti, DeiT-B, and LV-ViT-M (please refer to Table 13 in the PDF). Due to limitations in computational resources and time, we have temporarily not conducted experiments for LV-ViT-L; we will add its experimental results in the revised version. We greatly appreciate your suggestions.
>
> **Weakness 4**: We have supplemented some experimental results based on your feedback and that of other reviewers. We appreciate your suggestions.

---

> > ### Comment · Reviewer_qw7Z · 2023-08-11
> > **Response to Author's Rebuttal**
> >
> > I would like to thank the authors to respond to my comments.
> >
> > > **Efficiency gains**
> >
> > I appreciate the authors' efforts in providing throughput numbers. I would like to ask the following questions:
> >
> > A) How is the throughput measured (i.e. details of hardware, batch size, etc.) ?
> >
> > B) Recently, FasterViT [1] has been proposed as a strong contender with SOTA benchmarks on Top-1 vs throughput trade-off. I would like to ask if the authors can use FasterViT as a baseline method and apply their proposed technique ? it would be great to have throughput gains compared.
> >
> > The FasterViT [2] code is publicly published by their authors.
> >
> >
> > > **Hierarchical ViT**
> >
> > I respectfully disagree with the authors' notion of challenging the effectiveness of hierarchical ViT. Currently, all results are focused on the classification task. However for downstream tasks such as segmentation and detection, the multi-scale feature representation that are learned in hierarchical ViTs prove to be more useful and result in SOTA benchmarks. Please see various methods such as DINO [3] that use hierarchical ViT and achieve SOTA results.
> >
> > > **Threshold selection**
> >
> > I appreciate the authors' efforts in providing additional data in the attached PDF on the sensitivity of results to the threshold. As also acknowledged by authors, this still remains a weakness of the effort, as it would require somewhat an extensive grid-search to determine the right value. Nevertheless, it may be that the authors need to provide comprehensive guides for the users for commonly-used networks on how to determine the right value.
> >
> >
> > > **Hard-coded assumptions for head, middle, tail**
> >
> > The rebuttal still did not provide a justification of how head, middle and tail patchification scheme generalizes to all cases. And more importantly, if incorporating such priors is scalable to other tasks like segmentation, detection, etc.
> >
> > > **Large model experiments**
> >
> > I appreciate the authors' attention to this point. In fact in many practical use-cases, large models are typically used due to their higher accuracy.
> >
> > [1] Hatamizadeh, A., Heinrich, G., Yin, H., Tao, A., Alvarez, J.M., Kautz, J. and Molchanov, P., 2023. FasterViT: Fast Vision Transformers with Hierarchical Attention. arXiv preprint arXiv:2306.06189.
> >
> > [2] https://github.com/NVlabs/FasterViT
> >
> > [3] Zhang, H., Li, F., Liu, S., Zhang, L., Su, H., Zhu, J., Ni, L.M. and Shum, H.Y., 2022. Dino: Detr with improved denoising anchor boxes for end-to-end object detection. arXiv preprint arXiv:2203.03605.

---

> > > ### Author Response · Authors · 2023-08-15
> > >
> > > We sincerely appreciate your timely feedback, which has afforded us an opportunity for further discussion. Following several days of exploration, we are pleased to present the following response concerning the inquiries you raised.
> > >
> > > **Details of the throughput measurement.**
> > >
> > > >We performed throughput measurement on a single RTX 3090 with a batch size of 32, utilizing a dataset of 50000 images from the validation set of ImageNet. The total inference time was recorded, and the throughput was calculated by dividing 50000 by the total inference time.
> > >
> > > **Apply multi-granularity strategy to FasterViT**
> > >
> > > >To be honest, following a series of attempts, we regretfully acknowledge the inherent difficulty in applying the multi-granularity strategy to FasterViT. The reason behind this difficulty is that FasterViT is a Well-designed hybrid architecture combining Transformer and CNN. The multi-granularity strategy is proposed based on analyzing the plain ViTs. Our attempts to integrate the multi-granularity strategy inadvertently disrupt the framework of FasterViT, which is a precise structure that enhances throughput by balancing math-limited and memory-limited operations across its constituent modules. This undermines the original design intent of FasterViT. For more analysis in combination with FasterViT, you may kindly consider referring to the next comment.
> > >
> > > **Hierarchical ViT and Plain ViT**
> > >
> > > >We think that your viewpoint is also reasonable. As for detection tasks, the DINO, which you mentioned is built upon the Hierarchical ViT, continues to remain the most competitively poised model. Conversely, our MG-ViT, built upon the Plain ViT, outperforms models built upon the Hierarchical ViT for classification tasks. Furthermore, as for segmentation tasks, exemplified by this June release of SegViTv2[r26], which is also built upon the Plain ViT, achieves SOTA results. Regarding the question of which ViT structure is superior, we believe that it necessitates deliberation in the context of varying tasks and distinct constraints. Naturally, we acknowledge that most of the current and competitive methods are bulit upon the Hierarchical ViT.
> > >
> > > **Threshold selection**
> > >
> > > >We have taken your suggestion into careful consideration and deem it indeed valuable. As a result, we intend to create comprehensive guidelines to aid users in determining suitable values. These guidelines will be presented in the revised version. Thanks!
> > >
> > > **Multi-granularity patchification scheme**
> > >
> > > >We deeply apologize for our confusion, but we still lack a clear understanding of "all cases." I would appreciate it if you could kindly provide a more detailed explanation of your concern?
> > >
> > > >As for whether priors from multi-granularity patchification scheme can be applied to detection or segmentation, we must responsibly admit that we cannot provide a definitive answer at this time. Because we have yet to develop a comprehensive framework for seamlessly integrating the multi-granularity strategy into detection or segmentation tasks. Similar to our previous response, this topic will be explored in our subsequent research. However, we infer that the applicability is plausible. For instance, as for detection, considering that classification serves as the foundation for detection. The multi-granularity strategy, through the identification of "important regions," might potentially provide improved guidance for detection.
> > >
> > > **Large model experiments**
> > >
> > > >We appreciate your suggestions and recognition. Making the model more computationally resource-friendly has been a direction to which we have been committed.
> > >
> > > Once again, we extend our sincere gratitude for your timely response, which is indeed precious among numerous reviewers. Furthermore, you have generously provided us with many new insights! We eagerly look forward to further communication and discussions. Thank you!
> > >
> > >
> > > [r26] Zhang B, et al. SegViTv2: Exploring Efficient and Continual Semantic Segmentation with Plain Vision Transformers. arXiv:2306.06289, 2023.

---

> > > ### Author Response · Authors · 2023-08-15
> > > **More analysis in combination with FasterViT**
> > >
> > > However, we found that the Hierarchical Attention of FasterViT is quite novel. Because it enables a window attention-based architecture to achieve both global and local information interaction without the need of shifted window and patch merging, as required by approaches like swin-T. Therefore, we integrated Hierarchical Attention into our framework. In the structure of combining multi-granularity strategy and Swin-T (please refer to the response to Weakness 1 from Reviewer V3HF), we replaced the SW-MSA (Shifted Window Multi-head Self Attention) and token merging operations in Swin-T with the Hierarchical Attention. The following presents our experimental results. ($\epsilon=1$)
> > >
> > > | Method | Acc.(%) | FLOPs(G) | TP(img./s) |
> > > | - | :-: | :-: | :-: |
> > > | MG-ViT(w/o PPSM) | 81.1 | 4.3 | 1435 |
> > > | MG-ViT(DeiT-S) | 81.0 | 3.9 |1591 |
> > > | MG-ViT(Swin-T) | 80.8 | 4.7 | 1284 |
> > > | MG-ViT(Hierarchical Attention) | 81.0 | 4.1 | 1786 |
> > >
> > > We can find that the framework utilizing Hierarchical Attention achieves a better balance between computation and performance. This is because Hierarchical Attention enables more interaction with global information while constraining attention calculation in a window. Thank you very much for providing inspiring insights. Due to the limited time for author-reviewer discussion, moreover, FasterViT was released in June this year, so we are unable to explore furtherly in the short term. However, we are willing to consider it as a very valuable interesting for future research. Thanks again!

---

> > > > ### Comment · Reviewer_qw7Z · 2023-08-15
> > > > **Response to Author's**
> > > >
> > > > I would like to thank the authors for taking the time to conduct additional analysis and also reflect on the raised suggested points.
> > > >
> > > > In case of acceptance of this work, I would like to ask the authors to confirm if they would commit to the following:
> > > >
> > > > (1) Release the source code and all trained models along with a comprehensive guide on threshold selections for popular models.
> > > >
> > > > (2) Include the above table ( HAT vs Swin vs DeiT) as part of the **main text** and dedicate a section to dive into differences regarding hierarchical vs plain ViT.
> > > >
> > > > (3) Add support and pre-trained models for hierarchical models such as **Swin** and **FasterViT** in the published repository. Given the throughput gain as presented in the rebuttal, it only makes sense to add these models.
> > > >
> > > > (4) Expand Table 3 in the main text to include more results from Table 17 (especially those with better performance)

---

> > > > > ### Author Response · Authors · 2023-08-17
> > > > >
> > > > > I would like to thank you for your response. In light of your concerns, we hereby make the following commitments.
> > > > >
> > > > > 1. As we committed during the submission of our manuscript, the code of this paper will be made available after the paper is accepted. We acknowledge the necessity of providing a comprehensive guide on threshold selection. Consequently, we will set out to design and compose this guideline, which will be presented in the final version.
> > > > >
> > > > > 2. Thanks for your suggestions and guidance, which have led us to extensive cerebration and discussion. The outcomes are valuable. Consequently, in the final version, we are willing to relocate the ablation experiments to the appendix and present an exploration and discussion on the differences between hierarchical ViT and plain ViT in the main text. We believe that this perfection will also provide some inspiration to researchers in the community.
> > > > >
> > > > > 3. We committed that we will release support and pre-trained models for hierarchical models such as Swin-T and FasterViT. Due to the limited time for rebuttal and discussion, Our current design may exhibit faultiness. We will continue to refine the design of the multi-granularity strategy applied to hierarchical ViT to ensure that the eventual release is of higher quality.
> > > > >
> > > > > 4. We appreciate your review of our responses to the other reviewers. Just as promised to other reviewers, in the final version, we will present a completed Table3 to provide a more extensive comparison of model performance.
> > > > >
> > > > > We hope that our response has addressed your concerns. If you still have any concerns, we eagerly look forward to your feedback, as it would greatly contribute to the enhancement of our paper. If it would be possible, we would sincerely appreciate your reconsideration of the rating towards further support.

---

> > > > > > ### Comment · Reviewer_qw7Z · 2023-08-17
> > > > > > **Final Decision**
> > > > > >
> > > > > > My concerns have been sufficiently addressed by the author's rebuttal, and considering the commitments made by the authors, I increase my rating to **weak accept**.

---

> > > > > > > ### Author Response · Authors · 2023-08-18
> > > > > > >
> > > > > > > We greatly appreciate for your constructive feedback and meticulous review. And it is good to know that our response has helped address your concerns. Lastly, we would like to express our sincere gratitude for your increased rating and further support towards our work! We hope that this paper achieves satisfactory results, not in vain of your efforts and suggestions.
> > > > > > >
> > > > > > > Best wishes,
> > > > > > >
> > > > > > > All authors of Paper 6032.

---

### Author Rebuttal · Authors · 2023-08-09

We sincerely thank all reviewers and ACs for their time and efforts. We are encouraged that the reviewers approve our motivation or idea (qw7Z J3bo V3HF cCy5 L7yG), consider our method novel (J3bo V3HF), and recognize our presentation (qw7Z J3bo V3HF L7yG). Meanwhile, we deeply value reviewers’ precious suggestions and questions. Concerning common aspects, I want to address and clarify them here once again.

In this work, our focus is on compressing ViT rather than proposing a new, general ViT structure. Drawing inspiration from three phenomena observed during the classification process, we contemplate how to effectively integrate the multi-granularity strategy with common techniques to solve the problem. It is within the impetus of this motivation that the MG-ViT framework is conceived. Similar to classical papers in ViT compression, which mainly evaluated classification tasks, we have evaluated the performance of MG-ViT in classification and devised numerous ablation experiments to substantiate the rationale and efficacy of our compression approachSimple but work.

Reference\
[r1] DynamicViT: Efficient Vision Transformers with Dynamic Token Sparsification. NIPS2021.\
[r2] Patch Slimming for Efficient Vision Transformers. CVPR2022.\
[r3] Evo-ViT: Slow-Fast Token Evolution for Dynamic Vision Transformer. AAAI2022.\
[r4] Self-slimmed Vision Transformer. ECCV2022.\
[r5] AdaViT: Adaptive Vision Transformers for Efficient Image Recognition. CVPR2022.\
[r6] Not All Patches are What You Need: Expediting Vision Transformers via Token Reorganizations. ICLR2022.\
[r7] ATS: Adaptive Token Sampling For Efficient Vision Transformers. ECCV2022.\
[r8] Exploring Plain Vision Transformer Backbones for Object Detection. arXiv:2203.16527v2.\
[r9] MixMAE: Mixed and masked autoencoder for efficient pretraining of hierarchical vision transformers. CVPR2023.\
[r10] Convnext v2: Co-designing and scaling convnets with masked autoencoders. CVPR2023.\
[r11] Scale-Aware Modulation Meet Transformer. ICCV2023.\
[r12] Internimage: Exploring large-scale vision foundation models with deformable convolutions. CVPR2023.\
[r13] More convnets in the 2020s: Scaling up kernels beyond 51x51 using sparsity. ICLR2023.\
[r14] Global context vision transformers. ICML2023.\
[r15] What Limits the Performance of Local Self-attention? IJCV2023.\
[r16] DeepMAD: Mathematical Architecture Design for Deep Convolutional Neural Network. CVPR2023.\
[r17] Dilateformer: Multi-scale dilated transformer for visual recognition. IEEE TMM2023.\
[r18] PVG: Progressive Vision Graph for Vision Recognition. ACM MM2023 arXiv:2308.00574.\
[r19] BViT: Broad Attention-Based Vision Transformer. IEEE TNNLS2023.\
[r20] Conformer: Local features coupling global representations for recognition and detection. IEEE TPAMI2023.\
[r21] FLatten Transformer: Vision Transformer using Focused Linear Attention. ICCV2023 arXiv:2308.00442.\
[r22] Slide-Transformer: Hierarchical Vision Transformer with Local Self-Attention. CVPR2023.\
[r23] Joint Token Pruning and Squeezing Towards More Aggressive Compression of Vision Transformers. CVPR2023.\
[r24] Making Vision Transformers Efficient from A Token Sparsification View. CVPR2023.\
[r25] A Closer Look at Self-Supervised Lightweight Vision Transformers. ICML2023.

---

### Decision · Program_Chairs · 2023-09-21

**Decision:**

Accept (poster)

**Comment:**

After discussion, this submission received 4 positive scores and 1 negative score. After reading the paper, the review comments and the rebuttal, the AC think the major concern is about the lack of experiments for downstream tasks, which is encouraged to added to the camera-ready version. The AC discussed with the SAC, and he agreed the decision.